# Continuous lateral oscillations as a core mechanism for taxis in *Drosophila* larvae

Antoine Wystrach[1,2†], Konstantinos Lagogiannis[1*†], Barbara Webb[1]

[1]School of Informatics, University of Edinburgh, Edinburgh, United Kingdom; [2]Centre de recherche sur la cognition animal, CNRS, Universite de Toulouse, Toulouse, United Kingdom

**Abstract** Taxis behaviour in *Drosophila* larva is thought to consist of distinct control mechanisms triggering specific actions. Here, we support a simpler hypothesis: that taxis results from direct sensory modulation of continuous lateral oscillations of the anterior body, sparing the need for 'action selection'. Our analysis of larvae motion reveals a rhythmic, continuous lateral oscillation of the anterior body, encompassing all head-sweeps, small or large, without breaking the oscillatory rhythm. Further, we show that an agent-model that embeds this hypothesis reproduces a surprising number of taxis signatures observed in larvae. Also, by coupling the sensory input to a neural oscillator in continuous time, we show that the mechanism is robust and biologically plausible. The mechanism provides a simple architecture for combining information across modalities, and explaining how learnt associations modulate taxis. We discuss the results in the light of larval neural circuitry and make testable predictions.

**\*For correspondence:** klagogia@inf.ed.ac.uk

[†]These authors contributed equally to this work

**Competing interests:** The authors declare that no competing interests exist.

## Introduction

The larvae of *Drosophila* display taxis behaviours by spontaneously crawling towards or away from the source of stimuli such as odours, or more generally, up or down stimulus gradients, including chemical, light and temperature gradients (*Luo et al., 2010*; *Gomez-Marin et al., 2011*; *Gomez-Marin and Louis, 2012*; *Gomez-Marin and Louis, 2014*; *Kane et al., 2013*; *Klein et al., 2015*). This behavioural tendency is flexible and can be altered by associative learning if the stimulus is presented together with a positive or negative reinforcer (*Scherer et al., 2003*; *Gerber et al., 2004*; *Ache and Young, 2005*; *Diegelmann et al., 2013*; *Schleyer et al., 2015a*). The development of both a rich genetic manipulation toolkit and sophisticated behavioural assays have provided the basis for a recent explosion of studies targeting the biological underpinnings of larval taxis, as an ideal model system for investigating the neural basis of sensorimotor control and learning.

Larval chemotaxis, in particular, has been extensively studied. The main chemosensory organ is located on the head, and the small spatial separation of the bilateral olfactory receptors makes it unlikely that the animal can detect the instantaneous odour gradient. In fact, it has been shown that larvae can still chemotax with a single active receptor (*Fishilevich et al., 2005*; *Gomez-Marin et al., 2010*; *Louis et al., 2008*). The key information used by the larva appears to be the change in odour concentration experienced as it moves forward and/or casts its head sideways (*Gomez-Marin et al., 2010*). Olfactory sensory neurons are well suited to carry this information as they have been shown to give strong transient responses during changes in odour concentration (*De Palo et al., 2013*; *Nagel and Wilson, 2011*; *Kim et al., 2011*; *Schulze et al., 2015*) and the frequency and direction of turns (large body bends leading to a new trajectory direction) appears correlated to decreases or increases in the perceived concentration (*Hernandez-Nunez et al., 2015*; *Schulze et al., 2015*). Other sensory modalities could in principle use spatially separated sensors to detect instantaneous gradients across the body to direct steering, but recent studies reveal substantial similarity in the

characteristics of larval taxis behaviour across different modalities (*Gepner et al., 2015*; *Bellmann et al., 2010*; *Lahiri et al., 2011*). This suggests it may be possible to provide a more general account that elucidates the nature of the sensory-motor transformation during all forms of taxis, and how multiple stimuli combine.

Several models have been designed to capture quantitatively the observed larval behaviour during its approach to an odour source (*Davies et al., 2015*; *Hernandez-Nunez et al., 2015*; *Schleyer et al., 2015a*; *Gepner et al., 2015*). These models typically assume the expression of taxis consists of multiple behavioural states with state transitions that are biased by sensory stimuli. In *Davies et al. (2015)*; a model closely based on the behavioural analyses in *Lahiri et al. (2011)*; *Gomez-Marin and Louis (2014)*; *Gomez-Marin et al. (2011)*; *Ohashi et al. (2014)* reproduces many characteristics of larval chemotaxis by combining three mechanisms: biased forward runs (weathervaning), increased probability to stop runs when odour concentration decreases (klinokinesis), and increased probability to resume running when a head cast is in a direction that increases the experienced odour concentration (klinotaxis). Each contributes to the improvement of odour taxis performance, and in theory, each could be individually modulated by sensory stimuli characteristics, context, other stimuli, or learning, in a manner that modifies the observed odour preferences. However, behavioural observation shows rather strong similarities in the behavioural modulations resulting from apparently unrelated conditions, such as odour-tastant associative learning and variation of stimulus concentration (*Schleyer et al., 2015a*), which simultaneously modulate both the klinokinetic and klinotactic responses (weathervaning was not assessed in this study). Also, a recent attempt to categorise larval behavioural states using an unsupervised method based on the animal's posture suggests the existence of a continuum rather than clear-cut categories (*Szigeti et al., 2015*).

It remains possible that the apparent repertoire of taxis behaviours seen in the larvae is, in fact, the result of a single underlying mechanism. In this paper, we take a bottom-up, synthetic approach (*Braitenberg, 1986*) to investigate whether a simpler sensorimotor control scheme can give rise to the observed phenomena of taxis. We combined a detailed observation of the larva's crawling motions with an agent-based simulation to explore the behaviours that can emerge from the interaction between brain, body, and environment.

Specifically, inspired by the description in larvae of frequent low amplitude head sweeps, which modulate run direction (*Gomez-Marin and Louis, 2014),* and the idea that closed-loop sensory modulation of an intrinsic motor pattern can be a particularly efficient neural mechanism for flexible behavioural control (*Izquierdo and Lockery, 2010*; *Kanzaki, 1996*; *Levi et al., 2005*; *Willis and Arbas, 1997b*), we investigated the hypothesis that taxis in larvae results from continuous anterior body oscillations modulated by immediate sensory inputs.

Our analysis reveals that larvae indeed display continuous anterior body oscillation. We show, in both a simple discrete-time model and a neural model in continuous time, that direct sensory modulation of oscillation amplitude is sufficient to reproduce many specific larval taxis signatures, without requiring specific parameter tuning to different conditions. Finally, we discuss the biological relevance of our proposed mechanism and how it could provide a simple and robust solution for combining information across modalities, or from learnt and innate pathways, to modulate taxis.

## Results

### Evidence for continuous anterior body oscillation in larvae

We used previously recorded tracks of 42 wild-type larvae performing innate chemotaxis (*Gomez-Marin et al., 2011*) to analyse the body-bend, the anterior body angular velocity and the forward speed. This reveals a continuous alternation between left and right turns, which appears most clearly in the angular velocity of the anterior part of the body (*Figure 1A* 'blue line'). Larvae are known to regularly stop their forward peristalsis motion and display large lateral head sweeps (*Gomez-Marin et al., 2011*). A closer look shows that these head sweeps do not seem to break the continuous alternation between left and right turns, i.e., if the larva's head was moving left before stopping the peristalsis motion, the first head sweep after stopping will be to the right, and vice versa (*Figure 1B* and *Figure 2A,B*). Thus, these head sweeps appear to be part of a continuous oscillation rather than individual motor actions triggered independently. Also, the distribution of body bending, anterior body angular velocity and acceleration, as well as the extent of each lateral oscillation of the

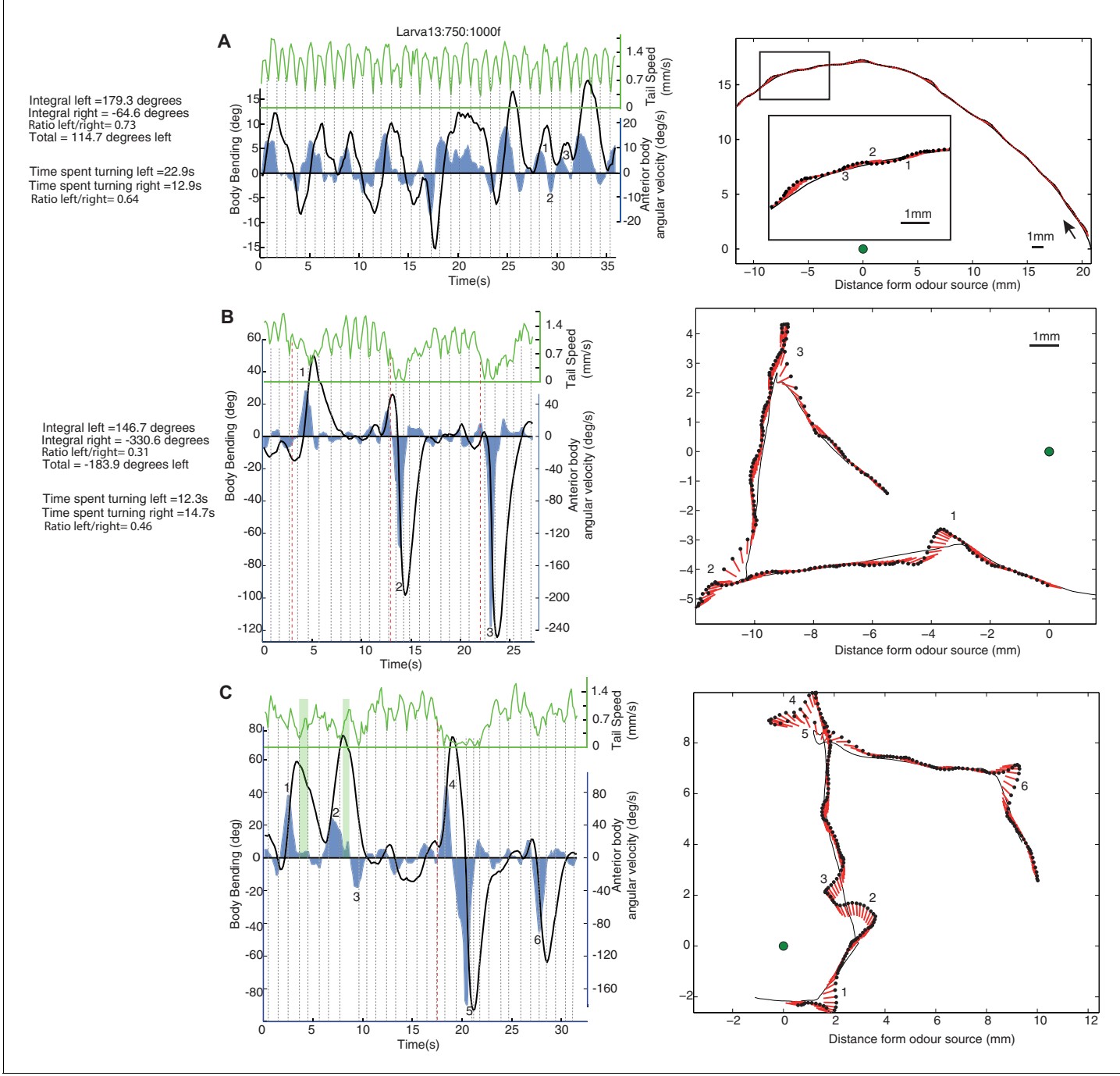

**Figure 1.** Examples of individual larva motion dynamics during chemotaxis show lateral oscillations. Left side panels show the angular speed of the anterior part of the body (blue), body bending (black), and peristaltic steps (grey dotted lines) based on tail speed (green) corresponding to the paths shown on the right. Events of interest are labeled by numbers. (A) Path section with no peristalsis inhibition. The larva shows a continuous alternation between left and right, but turning is biased in both amplitude and duration towards positive angles, resulting in a left curve. (B) Path section with an intermediate (1) and two stronger (2 and 3) peristalsis inhibition events that do not interrupt the turning alternation. (C) Path section with a peristalsis inhibition event covering two successive turns (4 and 5). The green vertical bars (1 and 2) indicate moments at which the body bending decreases (from left to right) even though the larva anterior body is still slightly swinging towards the left. This is due to the simultaneous forward peristalsis motion bringing the posterior part of the body towards the axis of the anterior part. The angular speed of the anterior body provides thus a better proxy than body bend to infer the control commands involved. (B,C) Red dotted lines indicate the onset of peristalsis inhibition (conservatively late measure) which occurs before any strong changes in angular speed or body bending.

The following figure supplement is available for figure 1:

*Figure 1 continued on next page*

*Figure 1 continued*

**Figure supplement 1.** Peristalsis and lateral oscillation rhythms appear decoupled.

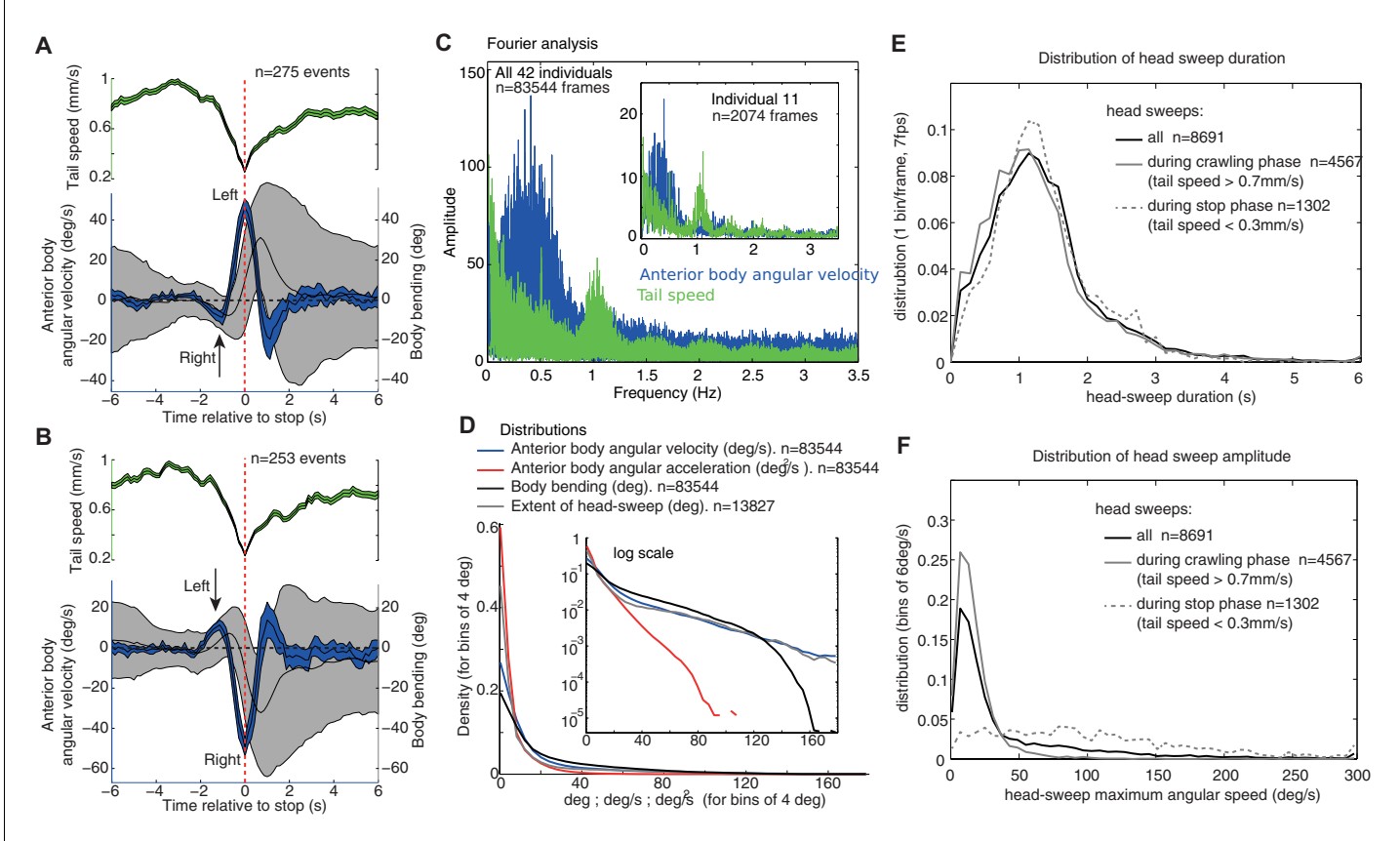

**Figure 2.** Group statistics of larva motion dynamics during chemotaxis support the existence of an intrinsic lateral oscillatory rhythm. (A,B) Average (±95%CI) dynamics of the anterior body angular velocity (blue), body bending (black) and tail velocities (green) displayed before and after the onset of peristalsis inhibition events aligned at $t = 0$ (dashed red line, when tail speed drops to a minimum), and categorised according to whether the larva is sweeping towards the left (A) or right (B) at $t = 0$. Drops in forward crawling speed tend to be accompanied by a large head sweep, as shown previously (*Hernandez-Nunez et al., 2015*) (see *Figure 2—figure supplement 1*). Large head sweeps are preceded by low-amplitude head casts in the opposite direction (arrows), suggesting the large head cast direction is dependent on the state of the oscillation. (C) Fourier analysis of the angular velocity of the anterior body (blue) and tail speed (green) across all larvae (see inset for a single individual). Tail speed (peristalsis) rhythm is fairly constant across larvae at around 1.0 Hz, with slight harmonics of the tail speed at 0.5 Hz, which results from the tendency of some larvae to alternate continuously between a weaker and stronger peristalsis wave (see *Figure 1A*'s tail speed for an example). The angular velocity of the anterior body (blue) shows a slower rhythm than the peristalsis, with a higher variation across and within individuals. Note that the rhythms are not multiples of each other, suggesting that they are operating independently (see also *Figure 1—figure supplement 1*). (D) Distributions of markers of the anterior body sweeps (see inset for logarithmic scale) reveal no sign of bimodality, suggesting a continuum of turning modulations rather than the triggering of distinct specific actions. (E,F) Distributions of individual head-sweep's duration (E) and maximum angular speed (F). Head sweeps are defined as the period between the two successive points in time where the anterior body angular speed crosses zero. (F) Head sweeps tend to reach higher angular velocity during stop phases (dash grey) than during crawling phases (grey) (i.e. when tail speed average during head sweep is <0.3 mm/s and >0.7 mm/ s, respectively). (E) Head sweep durations can vary; however, the distributions of duration are similar during stop phases (dash grey) and forward crawling phases (grey), suggesting a shared underlying oscillatory rhythm (E).

The following figure supplement is available for figure 2:

**Figure supplement 1.** Correlation of head sweep statistics to tail speed.

anterior body, show a smooth curve with no signs of bimodality (*Figure 2D*) suggesting a continuum of turning modulation rather than a discrete set of distinguishable actions.

A Fourier analysis confirms the existence of an oscillatory rhythm with a mean frequency around 0.3 Hz; that is roughly one turn left and one turn right every 3.3 s (*Figure 2C*, blue). This turning oscillation seems decoupled from the peristalsis motion (*Figure 1—figure supplement 1*), which operates around a mean frequency of 1.1 Hz (*Figure 2C*, green). The peristalsis rhythm appears remarkably constant, perhaps because of biomechanical constraints (*Ross et al., 2015*). Therefore, a direct coupling between peristalsis and turning oscillation would constrain the larvae to spend as much time sweeping left as sweeping right, which would restrict the flexibility in trajectory alterations. By having the lateral oscillations decoupled from peristalsis, however, the relative duration between left and right sweeps can vary. This is indeed what we observed in larvae. A curving path to the left for instance, is achieved by spending slightly more time (and also increasing the angular speed of the head sweep) sweeping left than sweeping right (*Figure 1A* 'blue region': Time spent turning right = 12.9 s; Time spent turning left = 22.9 s; Ratio right/left = 0.64. Integral left = 179.3 degrees; Integral right = −64.6 degrees; Total = 114.7 degrees left; Ratio right/left = 0.73). Eventhough larvae show larger and quicker head sweeps when the peristalsis motion has stopped (*Figure 2F*), the head sweep duration is similar between crawling phases and stop phases (*Figure 2E*), suggesting again the existence of a shared underlying oscillatory rhythm.

To summarise, our observations support the hypothesis that a continuous lateral oscillation of the anterior body sits at the core of the chemotaxis mechanism, and that its rhythm is decoupled to the peristaltic rhythm thus allowing more freedom to adjust the head-turning velocity and amplitude.

## Hypotheses and modelling assumptions

We embedded the idea that continuous lateral oscillation of the anterior body sit at the core of the taxis mechanism in two simple agent-based models, one running in discrete and the other in continuous time. Our hypotheses are:

- 'Small amplitude head-casts' and 'large amplitude head-casts' (*Gomez-Marin and Louis, 2014*) are manifestations of a single underlying mechanism that continuously drives a lateral oscillation of the anterior body (head casts).
- The *direction* (left or right) of a head-cast at a given time-step is determined only by the current state of an intrinsic oscillator rather than the sensory input or its history, or an active choice by the larva to probe the environment.
- The *amplitude* of each of these alternating head-casts is continuously modulated by the stimulus perceived.

We sought to simplify our models as far as possible so as to establish the nature of the essential sensorimotor components that could underlie the emergence of chemotactic signatures observed in larvae. Our implemented models therefore also make the following simplifying assumptions:

- Stopping (inhibition of forward peristalsis) is not essential for taxis, except insofar as it aids reorientation by enabling larger turns or tighter curvature of paths. Hence, we neglect stops, and in our model, the agent is continuously stepping forward, even when displaying large turns. Note that we address the limits of this assumption, and how stopping could be introduced to the model, in the discussion.
- As the anterior body bearing determines the forward step direction in larvae, we assume it is the crucial variable for taxis, and not the actual bend of the body. Therefore, we limit our model to a single oriented point in space, representing the position of the larva as a whole along with its current bearing. The control mechanism then determines the trajectory of that point through space. This way of abstracting the larval trajectory has been previously used in biological analysis (*Louis et al., 2008*) and enables us to compare our model to larval trajectory statistics.

## A simple oscillatory agent reproduces taxis

We first embedded the above ideas into a discrete-time agent (see 'Materials and methods'). At each time-step, the point agent rotates on the spot (by an amount $\alpha$, see *Figure 3A*, grey arrow) and makes a step forward of a fixed size $\lambda$ = 1 mm in this new direction (*Figure 3A*).

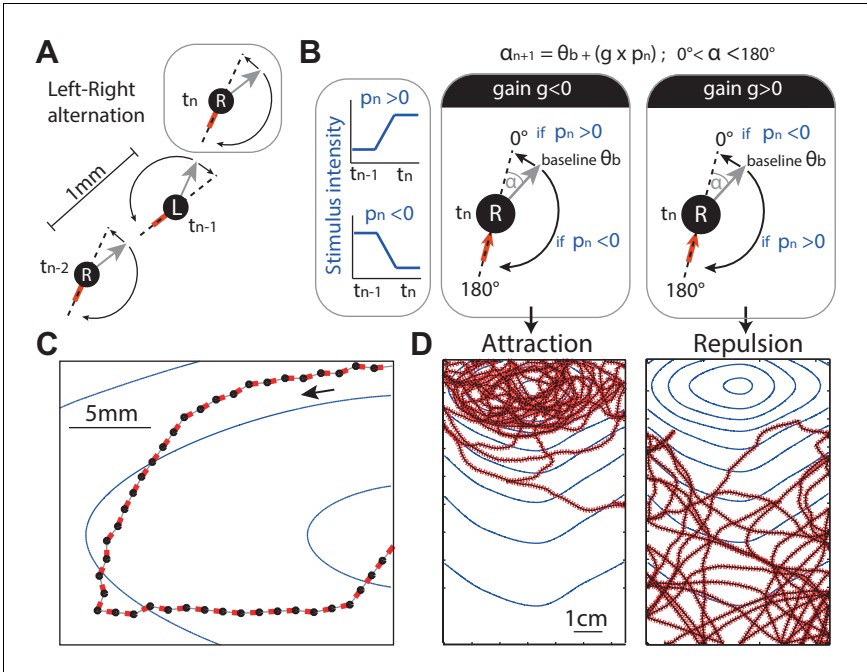

**Figure 3.** Discrete-time agent model. (**A**) The agent consists of an oriented point (black dot) from which the odour concentration is sampled, the grey arrow indicates its orientation and the red line indicates the motion trail. At each time step, the agent performs a rotation ($0 \leq \alpha \leq 180$ deg) on the spot alternating between left and right rotations ('L' and 'R') so as to set a new orientation (grey arrow), and then moves forward by 1 mm. (**B**) In the absence of stimulation, the extent of this rotation is set according to a constant baseline $\theta_B$ (typically $\theta_B$ =10 deg), which is alternated between L and R causing movement in a zig-zag fashion. In the presence of stimulation, the extent of the rotation (e.g., towards the right) is modulated by the change of stimulus intensity (blue line, $p_{n-1}$) perceived between the previous $t_{n-1}$ and current $t_n$ position. The modulation is simplified down to a linear constant gain $g$; so that the extent of the rotation (here towards the right) is: $\alpha = \theta_B + g \times p_{n-1}$. As a result, when $g<0$, an increase in stimulus intensity perceived ($p_{n-1}>0$) would decrease the extent of the rotation towards 0 degrees (i.e. the agent goes straight), whereas a decrease in stimulus intensity perceived ($p_{n-1}<0$), would increase the rotation up towards 180 degrees (i.e. agent makes a U-turn). Effectively, a negative gain ($g<0$) yields attraction towards higher stimulus intensity. Inversely, a positive gain ($g>0$) yields aversion (**D**) ($g = 0$ results in neither). (**C,D**) Section path examples from agent simulation in an odour gradient, with line colours as in (**A**). The underlying dark continuous line indicates the overall path taken by the agent. The blue lines indicate the isoclines of the odour concentration.

The following figure supplement is available for figure 3:

**Figure supplement 1.** Model's robustness to change in baseline angle.

The *direction* of these re-orientations alternates between left and right on each time-step (*Figure 3A*). This represents the continuous heading oscillation observed in larvae (*Figure 1*).

The *amplitude* ($\alpha$) of these left/right alternating re-orientations is bounded from above and below (*Equation 7*). For most results reported in this paper the lower bound is 0 degrees (prevents a 'right' turn becoming a 'left' turn or vice versa) and the upper bound 180 degrees to represent the maximum possible re-orientation given the larva body bending constraints (*Figure 3A,B* dashed line).

In the absence of any stimulus, the *amplitude* ($\alpha$) of these re-orientations has a baseline angle $\theta_B$. In the main results, we set $\theta_B$ = 10 deg, so as to roughly match the apparent small amplitude oscillations observed in larva *Figure 1A*. However, we show that the value of this parameter is not crucial for the emergence of taxis (*Figure 3—figure supplement 1*).

In the presence of stimulation, such as a gradient of odour concentration, the *amplitude* ($\alpha$) of each of these re-orientations is modulated by the stimulus perceived. The stimulus perceived is taken as the difference in stimulus intensity between the previous and current location (in our model: $p_n$)

(*Figure 3B*). The *amplitude* ($\alpha$) of the rotation is determined by a simple linear function: $p_{n-1}$ is multiplied by a constant gain $g$, and this is then added to the baseline intrinsic oscillations $\theta_B$ (*Figure 3B*). Thus, bearing angle can be bidirectionally modulated, that is, the signal perceived can lead to an increase or decrease in the amplitude of the next turn, as compared to the baseline angle $\theta_B$, depending on the sign of $p_{n-1}$ and the constant gain $g$ (*Figure 3B*).

The gain $g$ is taken to represent the sensorimotor transformation, which gives a linear relationship between perceived sensory stimulation and motor command. What this linear transformation could imply for the larva is considered in the discussion.

*Figure 3C,D* shows that this simple agent is sufficient for taxis to emerge. The behaviour is very robust to the choice of baseline turning angle $\theta_B$ or gain values (*Figure 3—figure supplement 1*). Effectively, a negative gain ($g<0$) yields attraction towards higher stimulus intensity because decreasing stimulation ($p_{n-1}<0$) triggers strong re-orientations, while increasing stimulation ($p_{n-1}>0$) straightens the path (*Figure 3B*). Inversely, a positive gain ($g>0$) yields aversion (*Figure 3D*), and a null gain ($g=0$) yields neither attraction nor repulsion. While the sign of the gain $g$ determines attraction or repulsion, the magnitude determines its strength: the larger the gain, the stronger the agent's reaction to the sensory stimulation is, and thus stronger attraction or aversion emerges in the resulting trajectories (see *Figure 3—figure supplement 1*, first row).

In the following sections, we examine the ability of this basic model to capture the typical chemotactic signatures observed in larvae, including path shapes, bearing to odour distribution shapes, sensory history, and their qualitative change resulting from typical manipulations such as change in stimulus concentration or associative learning.

## Characteristic taxis trajectories

An emergent property of our agent model is that for an attractive odour (i.e. a negative gain) the distribution of bearing angle to the odour's source shows two peaks around 90 and $-90$ degrees (*Figure 4*). Therefore, the agent tends to spend more time with the odour on its sides rather than directly in front or behind it. Interestingly, this is also true with real larvae (*Figure 4*).

For both larval and the agent generated paths, this tendency is emphasised while displaying no large turns (*Figure 4C* blue line) and large turns tend to happen while the odour is located behind (*Figure 4C* red line), a result consistent with previous findings (*Gomez-Marin et al., 2011*; *Schleyer et al., 2015b*). Spending time with the odour located 90 degrees on the side translates into orbiting around the odour source. This 'orbital behaviour' can be observed clearly in simulated trajectories from the deterministic (absence of random noise) version of our agent model (*Figures 4B* and *5B*).

However, in our models, increasing the gain (towards higher negative values) results in a qualitative change to the shape of agent's trajectories from circular orbits to those characterized by straight crossings over the odour source, and sharp re-orientation events once the peak has been passed over, that is, when the odour source is now located behind the agent (*Figure 4B*). When further away from the odour source, the perceived changes in concentration ($p_{n-1}$) are smaller so, as during orbital behaviour, the agent tends to spend time with the odour on its side. As a result, the model predicts different statistics depending on the proximity to the odour: when close to the odour, crossing-over paths occur, resulting in a flattening of the bearing-to-odour distribution curve. Examination of actual larvae paths reveals similar signature: crossing-over trajectories emerge when close to the odour source [except for the Or42a single receptor mutant larvae, that show an orbital behaviour (see below)], which indeed results in a flattening of the distribution curve (*Figure 4A,D*).

## Modulation of the chemotactic response as a simple change in gain

Chemotaxis in larvae can be altered in several ways. For instance, genetic alterations of the peripheral olfactory circuits that reduce the number of functional OSNs (*Louis et al., 2008*; *Gomez-Marin et al., 2011*) or receptor diversity (*Gomez-Marin et al., 2011*) increase the spread of trajectories around a source, and can produce trajectories that seem to maintain a distance from the source (*Figure 5A*).

A change in gain parameter $g$ of our model is sufficient to generate trajectories that capture these path signatures (*Figure 5B*). As the gain is reduced, the spread of trajectories increases, and orbital trajectories eventually arise: the agent maintains a radial distance from the source. Critically, these

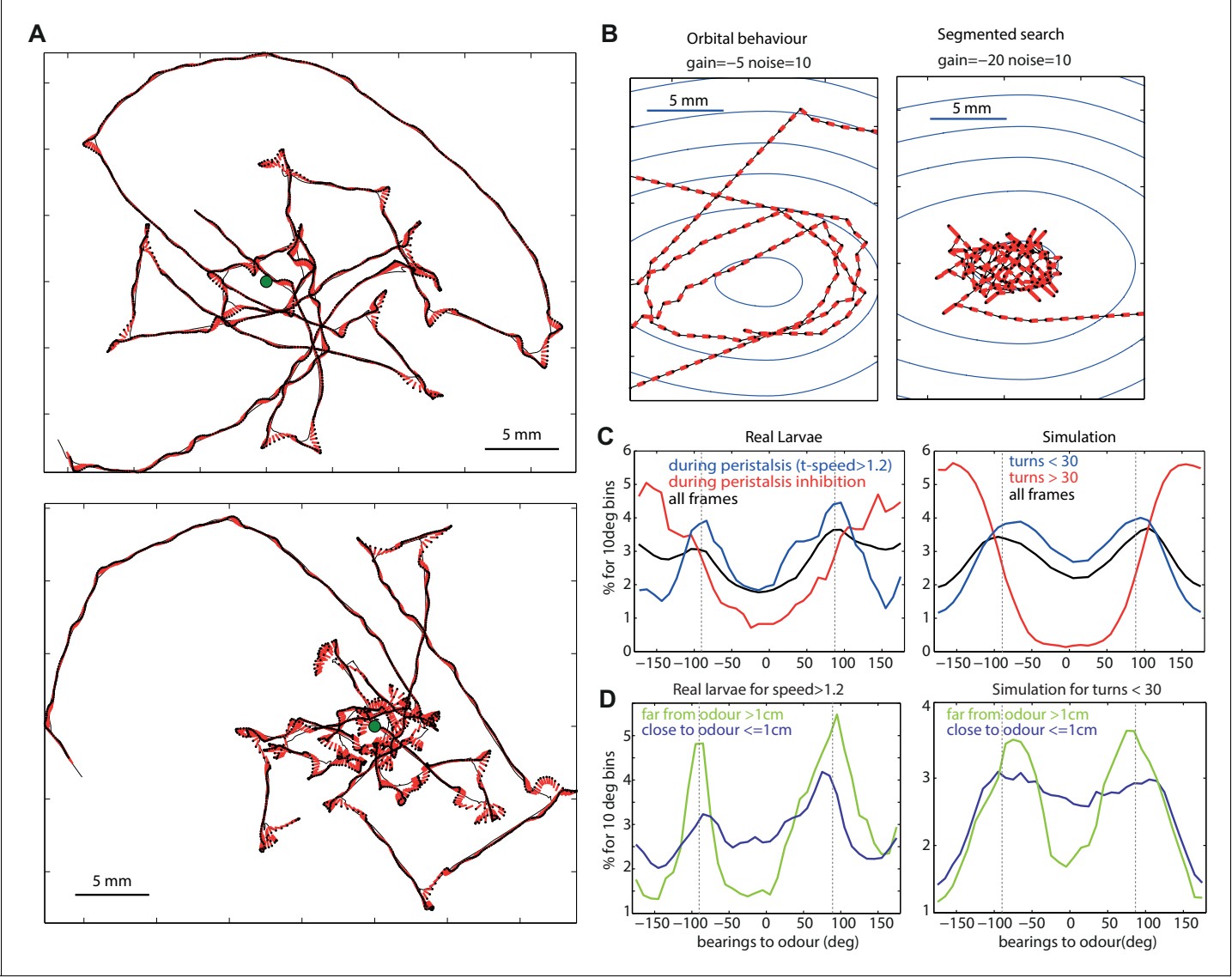

**Figure 4.** Typical path signatures for larvae and simulation. (A,B) Example of paths. (C,D) Distributions of bearings to odour. Both larvae and simulated agents tend to spend most time with the odour located on their sides (−90 and 90 degrees), orbiting the source. In both larvae and simulation, orbital behaviour is emphasized during peristalsis forward motion (turn < 30 degrees for the model) (C blue curve), and when the larvae/agent is more than 1cm away from the odour (D green curve). Crossing-over trajectories, by contrast, are constituted of regular large turns that happen mostly while the larvae/agent is heading away from the odour (C red curve) and is rather apparent when the larvae/agent is close to the odour (D blue curve).

are not due to active repulsion (*Gomez-Marin et al., 2011*; *Kreher et al., 2008*; *Asahina et al., 2009*) but due to a weak sensory-to-turn gain $g$ that does not allow the agent to perform sufficiently large reorientations to track the peak concentration as the agent moves in the arena.

*Schleyer et al. (2015b)* describe in detail the effects of changing the concentration of the odour source on the statistics of larval chemotaxis. Perhaps surprisingly, they show that similar behavioural effects are obtained after associative learning (*Figure 5C*). Here also, a change in stimulus intensity or a change in gain $g$ in our model closely captures these effects (*Figure 5D*). The explanation of why both kinds of changes yield similar effects is straightforward in our model as a change in stimuli intensity perceived $p_{n-1}$ or gain $g$ both directly affect the next head-sweep amplitude (see *Equation 6*).

A widely used summary measure of chemotaxis is the performance index: the proportion of a group of larva that is near the odour source after some period of time. In our model, the

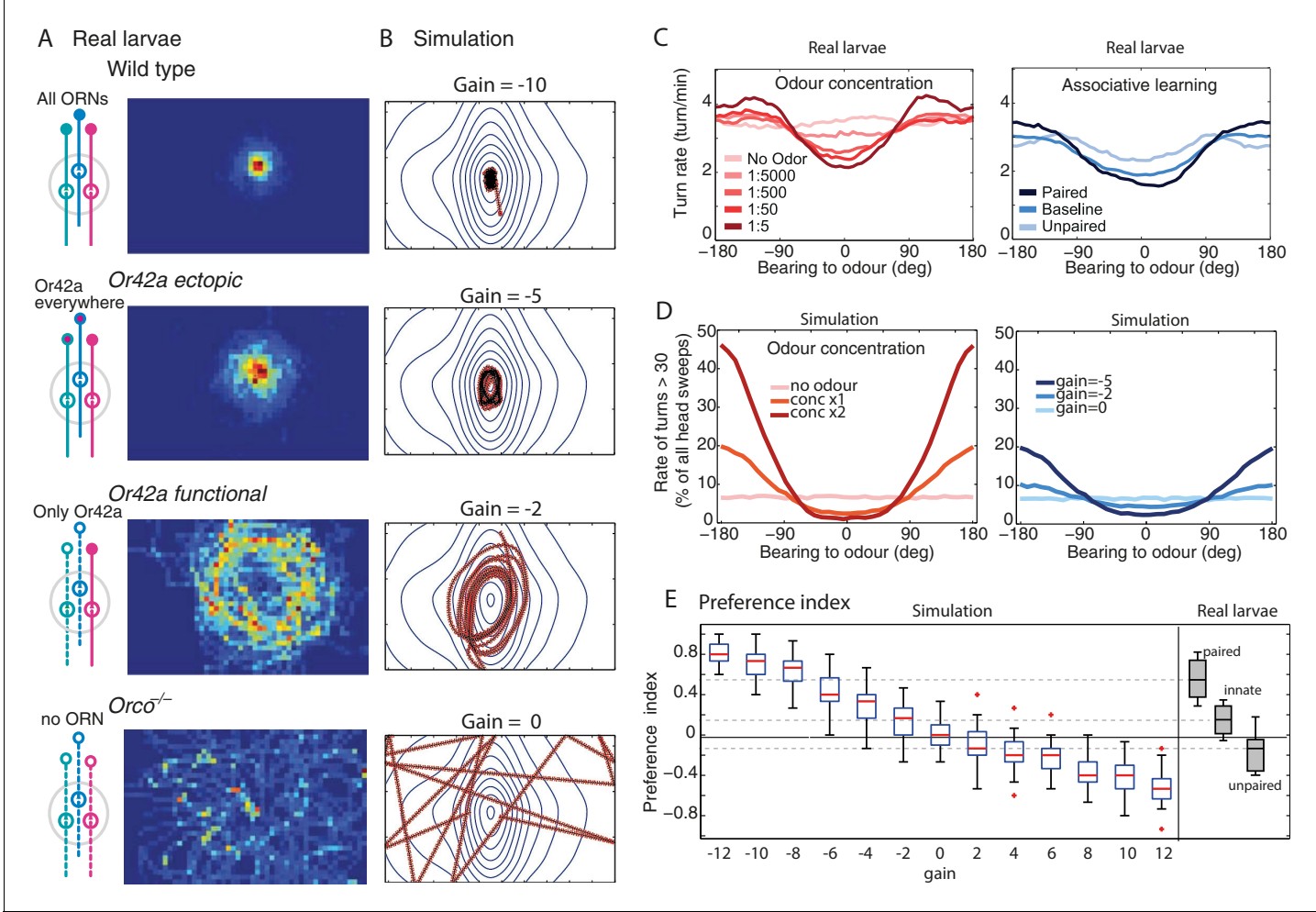

**Figure 5.** Modulation of the chemotactic response. (A) Time occupancy spatial maps for genotypes with re-engineered peripheral olfactory circuits tested in the near-source paradigm (30 mM odour source): wild type (N = 42 flies), Or42a ectopically expressed in the 21 intact ORNs (all neuron pairs active, N = 38), Or42a single-functional ORN (one pair of neuron active, N = 37), and Orco null (anosmic flies, N = 55) adapted from *Gomez-Marin et al. (2011)*. (B) The simulated agent can capture the patterns observed in larvae by changing *g*, suggesting that OSN activity acts collectively to increase the turning modulation signal. (C) Effect of odour concentration and appetitive conditioning on turn-rate (larva data from *Schleyer et al., 2015b*). (D) In our simulation (shows mean ± std. dev.), turning events were categorised as large turns if > 30 degrees and not followed by another large turn. Changes in stimulus intensity were obtained by multiplying the gradient by a factor 0, 1 or 2. Learning was modelled as a change in gain ($g = 0$, $g = -2$ or $g = -5$). The same qualitative changes in turn-angle and turn-rate relative to odour bearing are observed. (E) Preference index (($N_{odour-side} - N_{other-side})/N_{total}$) for 30 simulated larvae after 3 min, for different gains (larva data from *Schleyer et al., 2015b*).

performance index increases quite linearly with change in gain (*Figure 5E*) and can therefore account for the continuum of performance indices observed in larvae across experiments. This was not necessarily the case in previous agent simulation models, where small changes in parameter values would yield drastic changes in performance index (*Davies et al., 2015*).

It thus seems that our model can capture a variety of effects observed in path signatures, detailed motor changes and performance indices by changing in a single parameter (*g*). In the discussion, we reflect on possible implications of this result for the architecture underlying chemotaxis in larvae.

## Sensory history preceding turns

The average sensory history perceived before the occurrence of large turns shows a slow monotonic decrease in concentration which extends up to 10s prior to the large turn. This has been reported for larvae during chemotaxis (*Gomez-Marin et al., 2011*) or as a response to white noise

optogenetic stimulation of olfactory receptor (*Gepner et al., 2015*), and can also be observed in our model (*Figure 6*).

In larvae, this may suggest the existence of a low-pass filter enabling larvae to integrate monotonic decreases over relatively long time scales to increase the probability of triggering a large turn (*Gomez-Marin et al., 2011*; *Davies et al., 2015*; *Gepner et al., 2015*). However, our model does not possess such a low-pass filter: large turns occur as the consequence of the stimulus change perceived during the last time-step only. Here, the amplitude of a turn correlates with the size the stimulus change $p_{n-1}$, with maximum change occurring when the agent moves directly down-gradient. The re-orientation towards down-gradient causes a progressive decrease of the stimulus perceived,

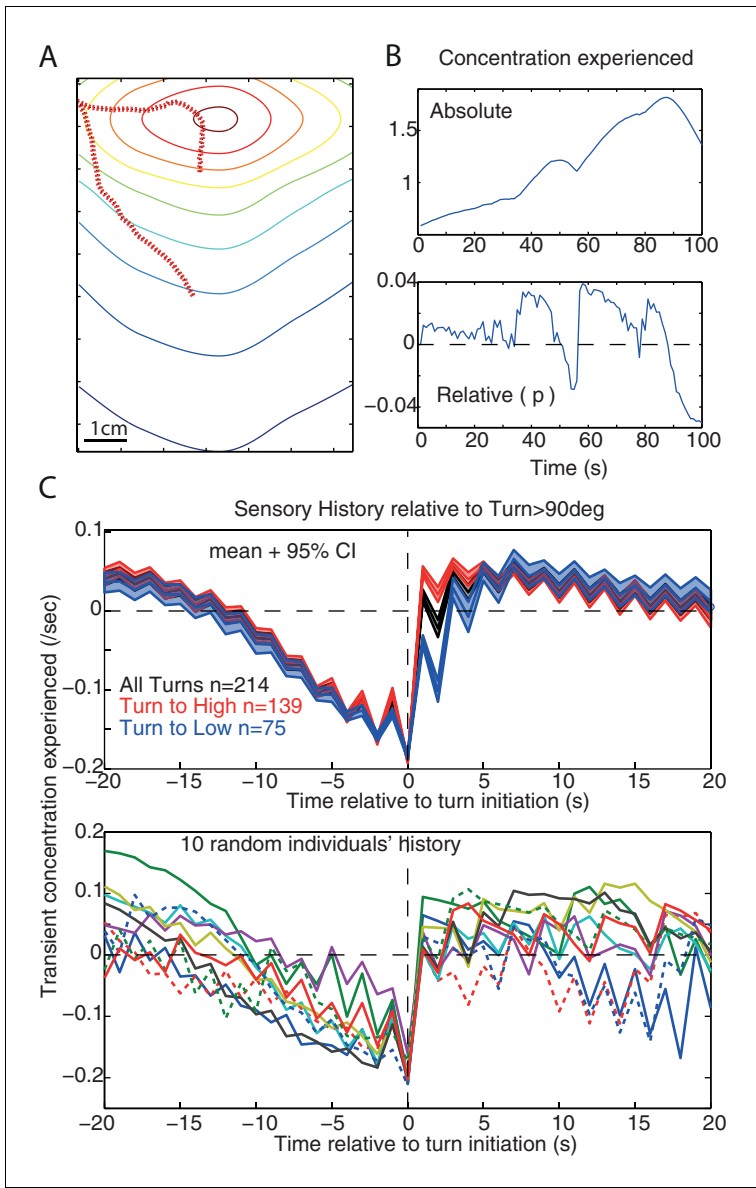

**Figure 6.** Sensory history and monotonic decrease in concentration. (A,B) Example of simulated path and the associated sensory history given the absolute ($s_n$) and relative ($p_n$) odour concentration perceived. (C) Average (±95%CI) and individual's example of the sensory history experienced before and after large turn events (>90 degrees) in our simulation, for all large turns (black), or only the large turns that result in experiencing a positive (red) or a negative $p_n$ (blue). A slow monotonic decrease in concentration precedes large turns even though turns are the consequence of the signal experienced during the previous step only.($g = -5$, noise = 10 deg).

which is reflected in the stimulus history. Since turns are more likely to be triggered when the larva faces away from the odour then averaging the stimulus around these turn events will reflect this history of monotonic sensory decrease. Therefore, the monotonic sensory decrease can result from the structure of the environment, and not from a low-pass filter in the larva.

However, we cannot give a sufficient explanation as for why the monotonic decrease is also seen under white noise optogenetic stimulation (*Gepner et al., 2015*). Our agent is a point in space, and therefore does not capture the details of head sweep movements through space. Further investigation of dynamics at this level would require implementing our agent into a more elaborate model that accounts for the larva's body, which promises to be an interesting endeavour. Nonetheless, the emergence of this monotonic decrease from the interaction between our reactive agent and the environment suggests caution is needed when interpreting the causal implications of sensory history prior to actions.

### First-turn bias

Larvae show a slight tendency to bias their first head cast (after a stop event) towards the side of the attractive stimulus (i.e. the odour side (*Gomez-Marin et al., 2011*; *Gomez-Marin and Louis, 2012*) or darker side during negative phototaxis (*Kane et al., 2013*), or towards preferred temperatures (*Luo et al., 2010*). This may suggest the involvement of bilateral sensing to obtain gradient information, or a memory of gradient information obtained during the run. But if we identify 'turns' in our model as those re-orientation angles exceeding the threshold that is usually associated with stopping in the larva, the agent also reveals a tendency to bias its first 'turns' towards the odour source (*Figure 7*), despite having no gradient information other than the change from one time step to the next. This tendency arises because of the oscillatory nature of the agent. Given an attractive odour (i.e. a negative gain $g<0$), large re-orientations are more likely to be triggered when a negative $p_{n-1}$ has been perceived during the previous step. And since a negative $p_{n-1}$ is more likely to be

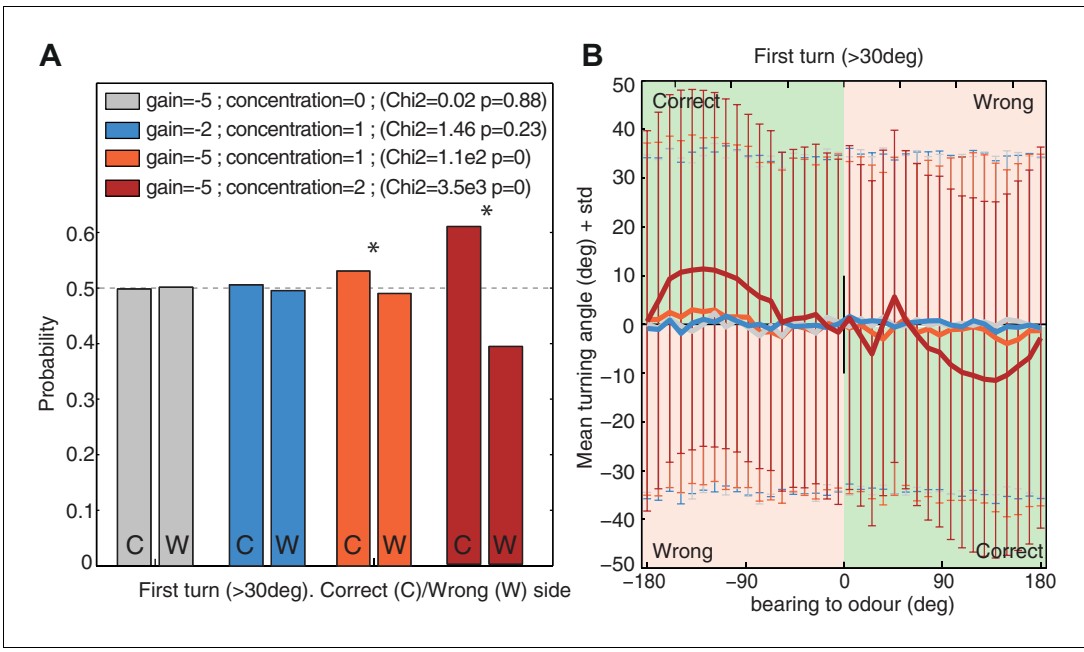

**Figure 7.** First turn bias towards the correct side. The model's first turns were categorised as large turns (>30 degrees) that were not preceded by a large turn in the previous timestep. (A) Probability of turning to the correct side (i.e. towards the odour; 1st column 'C') and wrong side (second column 'W'). Higher signals (i.e. increased concentration or stronger gain $g$) increase the bias. (B) Turning direction (mean ± standard deviation) given the bearing to odour. Green and red zones indicate turns towards (i.e. correct side) or away (i.e. wrong side) from the odour respectively. Noise $Z = 10$. Time $t = 800$. larvae ($n>25000$ for each group).

perceived when turning away from the odour, the subsequent turn, in the opposite direction, is thus more likely to be directed towards the odour side.

Nonetheless, this bias is weak and requires a large dataset to appear significant. Our model predicts that the bias should increase together with increasing odour attraction, whether from stronger gain due to appetitive learning; stronger sensory input due to increased odour concentration, or both (*Figure 7A*); it should also become more apparent when the odour is located on one side of the larvae (*Figure 7B*).

## A neural implementation of oscillation

So far, we have used a simple discrete time model to examine whether the basic principle of continuous lateral oscillations, modulated in amplitude by the stimuli perceived, can account for larval taxis. The discrete-time sensorimotor model assumed that head-sweeps and sensory perception are instantaneous and synchronized, which might imply that the descending sensory signal in the larvae may require precise timing in order to interact with the ongoing motor control of the oscillation.

Here, we aim to investigate whether our hypothesis can be verified in continuous time given the biophysical constraints of a neural implementation. The agent is again abstracted to an oriented point-sensor but now, critically, the change in heading is driven by a neural oscillator in continuous time, while sensory stimuli are continuously updated under the agent's motion through the environment.

For our purposes, we adapted (see 'Materials and methods' ) a spike-rate neural model of a central pattern generator (CPG) that has been successfully used to model lamprey locomotion (*Cohen et al., 1992*; *Lansner and Griller, 1997*) see *Marder and Calabrese, 1996*. The CPG consists of a pair of compartments, here taken to be driving the changes in the agent's heading (*Figure 8A*). Each compartment has a pool of self-connected excitatory neurons ($E$), and a cross-inhibitory interneuron ($C$) projecting to the other compartment. This produces a regular alternation in firing bursts between left and right sides that can be modified by the additional bilateral inputs, $A$ and $S$. The $A$ unit represents descending sensory signals either processed or direct, while the $S$ unit represents a modulatory signal. The spike rates from both compartments drive the changes in heading angle via a simple mechanical model (*Figure 8B*) , see 'Materials and methods'). Note this agent, like the discrete-time model, does not incorporate stops. It moves forward at a constant speed in the direction pointed by the heading angle. In the absence of stimuli arriving from the input unit $A$, the parameters of the system have been set so that it produces a regular ±10 degrees oscillation in the heading at around 0.3 Hz. The sensory input allows this oscillation to be perturbed, modifying the amplitude and phase relationships between the bursts of each side, resulting in a change in the agent's heading.

We evaluated the effect of stimulus timing against oscillator phase on the ability of this model to express overt bearing changes, by delivering perturbations to the input $A$ (note both sides $L$ and $R$ receive the same perturbation) at different points in the oscillation cycle. Measuring the overall change in bearing against a bilateral step-input of magnitude $A_m$ across different times $t_s$ showed that the larva can be steered in a direction determined by the sign of $A_m$ and, crucially the state of the oscillator at time $t_s$. The resulting steering varied smoothly across the oscillator phase and therefore it is not critically dependent on the precise timing of the perturbation (see *Figure 8F*).

In a virtual odour environment, the continuous agent also produces curved paths when further away from the odour source, characteristic of larval behaviour, that subsequently become orbits around the odour source (*Figure 8D*). The parameters of the model have been set such that the frequency of oscillation is within the ranges observed in larva (see *Figure 8C*), and thus when measuring the mean frequency of the heading velocity over such trajectories we obtain a noisy frequency spectrum comparable to the larval trajectory data. Further, we established that a doubling of the gain (from $g = 70$ to $g = 140$), which effectively doubles the input due to the sensory induced perturbations from the input $A$, results in a qualitatively comparable change from orbital to crossing-over trajectories, as observed with the discrete-time agent (*Figure 4*). The heading velocity dynamics emerging from the model (*Figure 8E*) are reminiscent of the experimental data of anterior body angular velocity (*Figure 1*) in terms that in both there is a baseline rhythm while heading velocity increases under larger re-orientations (see *Figure 1*).

The sensory stimuli during a change in heading naturally fall into a fixed phase relationship to the oscillator activity since the turn direction depends on which oscillator compartment is active.

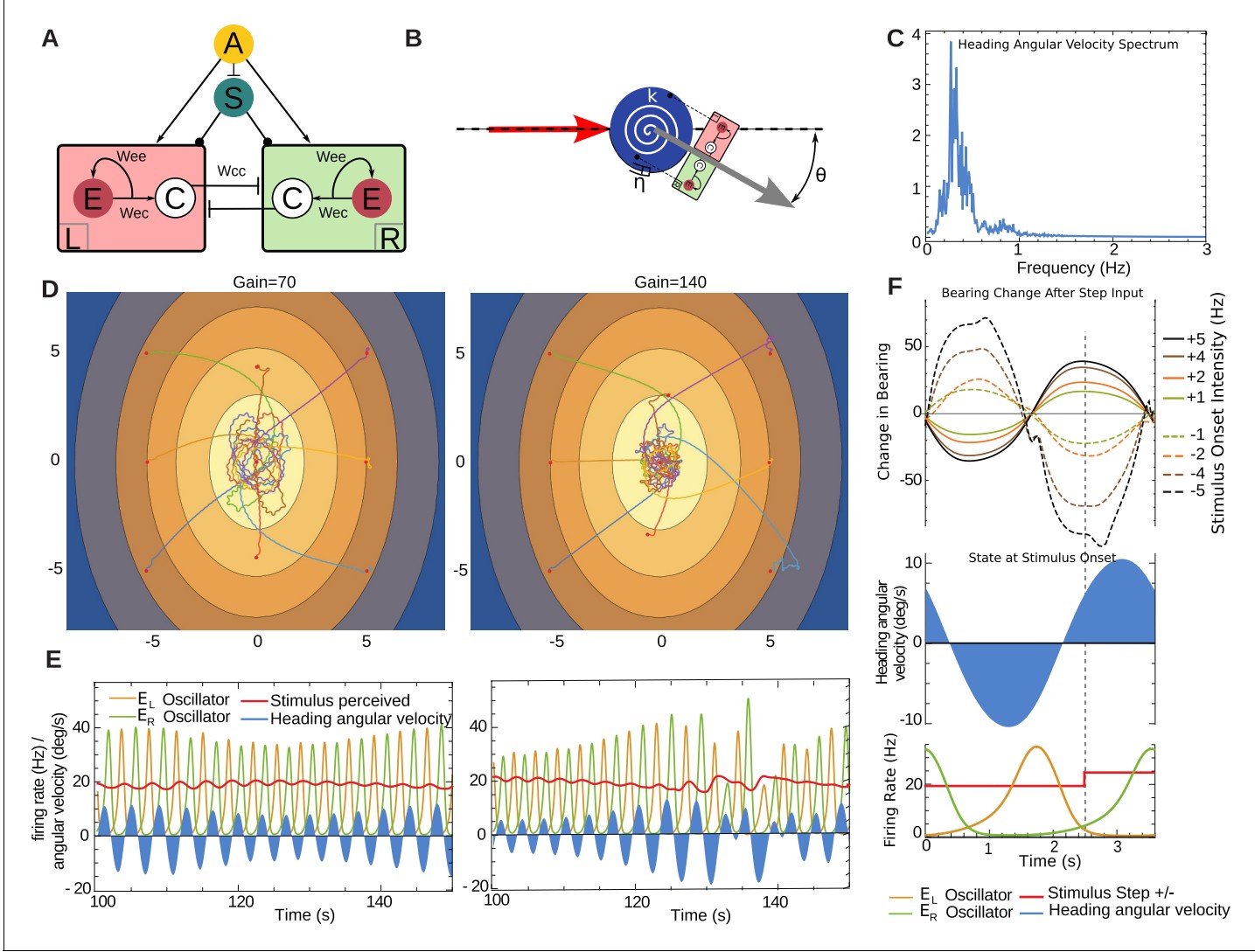

**Figure 8.** Neural model in continuous time. (A) Central pattern generator modelled with neurons of mean firing rates (adapted from *Wilson, 1999*). Arrows indicate excitatory connections, a bar denotes an inhibitory connection and circle denotes a neuromodulatory connection. Cross inhibitory connections go to all neurons of the opposite compartment. The $A$ unit represents mean firing rate of an OSN, and it projects to both compartments. The $S$ unit represents a neuromodulatory neuron which modulates the half-response threshold of the $E$ and $C$ neurons to effectively imitate the effects of a slow adapting current. We denote the left and right $E$ as $E_L$ and $E_R$ respectively. (B) A torsional spring is acting on the agent heading change to represent the restoring viscoelastic forces of the larva body bends. The red arrow indicates the direction from which the agent is coming, and the grey arrow indicates its heading direction. (C) Frequency spectrum of heading velocity oscillations that are comparable to larval data. (D) Example taxis trajectories in a virtual odour gradient with different gain. (E) Sample oscillator dynamics during chemotaxis, showing $E_L$ and $E_R$ alternating, while the $A$ input as influenced zig-zag motion of the agent in the environment. Under high-gain the turns appear sharper as required. (F) Effects of unit-step perturbations on bearing angle across oscillator phase. Panels below show the simultaneous state of the heading angular velocity, an example step-up in the firing rate of $A$ and the respective state of neural bursts from $E_L$ and $E_R$ (here shown unperturbed by the step-input $A$).

Therefore, the input perturbation perceived in the virtual odour environment relates to the turning motion, as these two variables are in a closed-loop. Consequently, increasing the gain also increases heading angular velocity and thus results in sharper re-orientation maneuvres, which in turn result in larger sensory perturbations.

## Discussion

### Oscillation as a principle of larval locomotion

Rhythmic behaviour is ubiquitous in biological systems. Producing oscillations during locomotion is widespread in biological systems, from multisegmented animals (*Iino and Yoshida, 2009*; *Izquierdo and Lockery, 2010*; *Lansner and Griller, 1997*) down to single cells (*Yang et al., 2011*), and may have advantages for sensorimotor control when tracking up an odour trail or plume (*Hangartner, 1969*; *Farkas and Shorey, 1972*; *Budick and Dickinson, 2006*; *Willis and Arbas, 1997a*; *Cardé and Willis, 2008*).

We have presented evidence from tracking of Drosophila larva that their locomotion also contains a rhythmic lateral oscillation (*Figures 1* and *2*), which is apparently uncorrelated with the peristaltic rhythm (*Figure 1*). We take this continuous oscillation to be the underlying basis for larval behaviours that are often treated as distinct states triggered by dedicated sensory motor processes (see *Green et al., 1983*; *Sawin et al., 1994*; *Cobb, 1999*; *Vogelstein et al., 2014*; *Ohashi et al., 2014*; *Gomez-Marin and Louis, 2012, 2014*; *Hernandez-Nunez et al., 2015*; *Gepner et al., 2015*). That is, we suggest running/weathervaning and casting/turning all result from the same underlying and continuously active oscillatory mechanism (*Figure 9C,D*), and that taxis involves continuous and direct sensory modulation of the oscillation amplitude. We show that models embodying this hypothesis, despite their simplicity, are sufficient to capture a range of taxis phenomena observed in larvae (*Figures 4*, *5*, *7*, *8*).

In this oscillatory taxis mechanism, 'directed' motion by the animal towards a target does not require a lateralised descending signal, nor does it include any switch between states or actions. Robust steering simply emerges from the closed-loop nature of the system: the oscillations control the exposure of the sensor to the stimuli, and the sensory signal controls the oscillations by perturbing the stable cyclic dynamics of the oscillator. A CPG is believed to operate within the thoracic and abdominal segments of larvae, executing a motor program for exploratory locomotion (*Hughes and Thomas, 2007*; *Berni et al., 2012*; *Lemon et al., 2015*), while a recurrent CPG circuit generating oscillations is consistent with cross-connections in the ventral nerve cord of the larva (*Kohsaka et al., 2012*; *Rickert et al., 2011*). Indeed, genetic disruption of the mid-line connection pattern, particularly in the anterior segments (T1, T2, T3), disrupts lateral body bending (*Berni, 2015*). This requirement echoes the essential neural architecture to implement our hypothesized oscillator.

### A simple sensorimotor mapping

We propose that the key mapping underlying taxis behaviour is a direct relationship of the perceived sensory signal to the modulation of oscillation amplitude. This direct relationship can thus be modelled as a single parameter: the gain $g$. In our models, both the valence (attractive or aversive) and salience (strength of attraction or aversion) of the oriented response along a gradient of stimulus intensity are determined by the value of $g$. For instance, in our abstract discrete-time agent, a high negative gain leads to a strong attraction, whereas a low but positive gain will lead to a moderate aversion (*Figure 5C*).

This 'gain' is only a model parameter and does not represent any specific neuro-anatomical feature in larvae. Rather, it could be seen as the net effect of the complete pathway from the stimulation of sensory receptors to muscle contraction. However, the fact that a single degree of freedom is sufficient to capture a substantial range of characteristics of the animal's behaviour, including the path signatures (*Figures 4*) as well as the effect of odour-tastant associations, or reductions in the number of ORs (*Figure 5*) can be informative for reconstructing the neural architecture underlying chemotaxis in larvae.

A simple picture emerges as to how larvae may modulate their chemotactic response, whether as a result of habituation (*Cobb and Domain, 2000*; *Larkin et al., 2010*), learning (*Ache and Young, 2005*; *Scherer et al., 2003*; *Gerber et al., 2004*; *Diegelmann et al., 2013*; *Schleyer et al., 2015a*), motivation (*Wang et al., 2013*), developmental maturation (*Gong et al., 2010*; *Krashes et al., 2009*) or by integrating multiple stimuli. All parallel sensory pathways simply need to converge and sum their signals (*Figure 9C*), whether inhibitory or excitatory. The net sum will determine the valence (attraction or repulsion) and the strength of the chemotactic response.

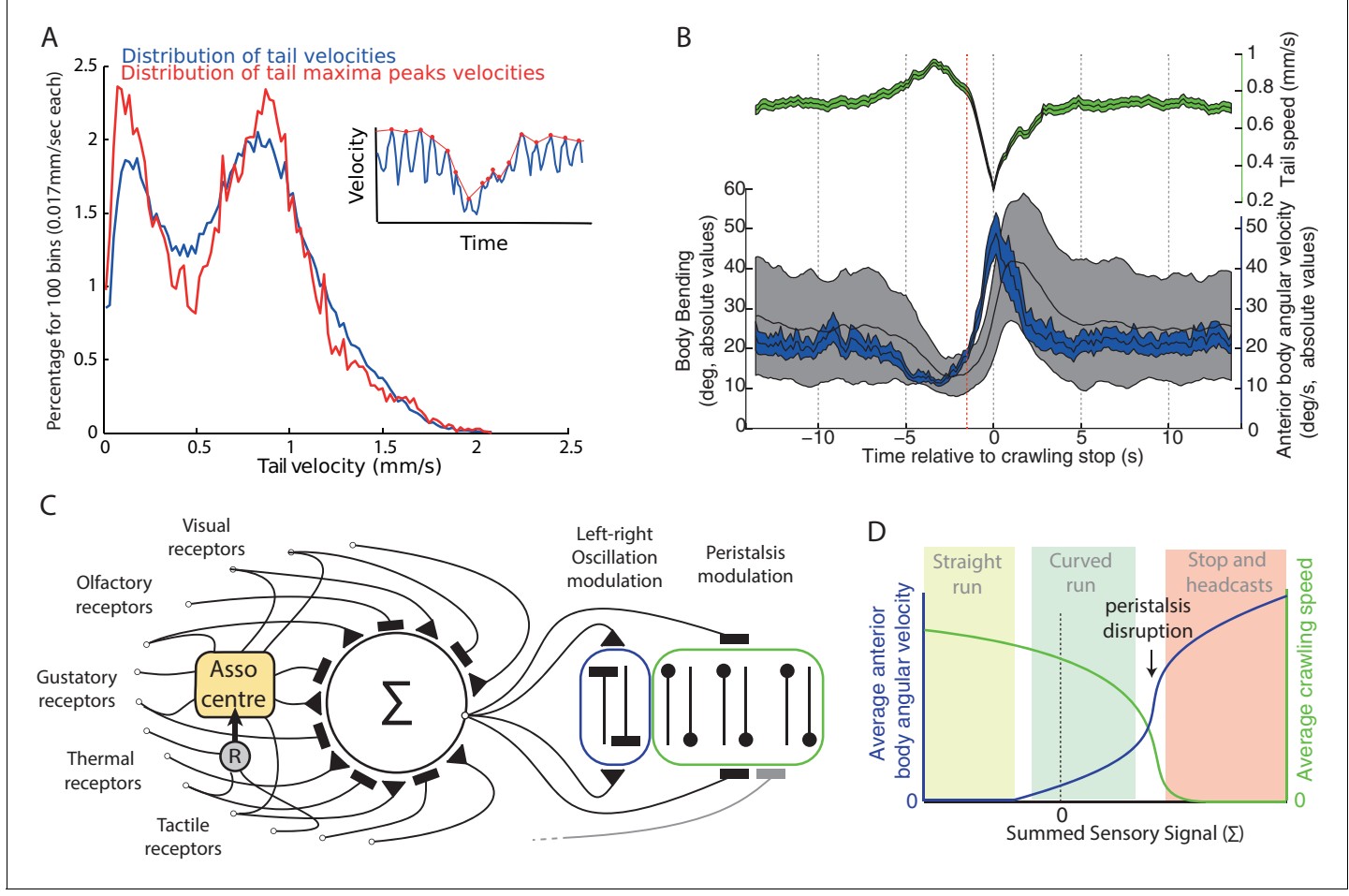

**Figure 9.** Oscillation and peristalsis modulation. (**A**) Real larvae tail velocities show a bimodal distribution, with the first peak corresponding to peristalsis inhibition events. Inset illustrates the extraction of maxima peaks (red curve) of tail velocities. (**B**) Average (±95%CI) of the tail velocities (green) and absolute values for the anterior body angular velocity (blue) and body bending (black) displayed before and after peristalsis inhibition events (aligned at t = 0, when tail speed dropped to a minimum). Red line highlights that average tail speed velocity (green) starts dropping before the occurrence of a large increase in body bending (black) or anterior body angular velocity (blue). This suggests that tail speed is not a mere physical consequence of a large turning event. (**C**) Conceptual scheme illustrating our overall view: all modalities, innate and learnt, are integrated at the zone of convergence. The summed signal is sent to both the neural oscillator mediating turning of the anterior body (blue), and to neurons mediating peristalsis inhibition (green). Associative center (Asso center), such as the mushroom body, where signal weightings can be modulated given the co-activation of a reinforcer neuron (R). Note that our scheme allows for sensory inputs to bypass the zone of convergence, and send their signals directly onto local control of the oscillator and/or the peristalsis motion. (**D**) Qualitative depiction of how apparently distinct behaviours could emerge from a continuous modulation signal. Growing signal strength increases anterior body angular velocity (blue) and inhibits crawling speed (green) simultaneously. If the signal is sufficiently strong peristalsis disruption happens (arrow), leading to an abrupt drop of tail speed velocities. The relaxing of the peristaltic synchronous left-right body contraction enhances the reaction to the thoracic left-right asynchronous oscillatory contraction, thus resulting in sharp increase in head sweep amplitude. Peristalsis spontaneously resumes when the sensory command lowers below peristalsis disruption threshold. The combinations of these two parameters (blue and green) can lead to the emergence of straight runs (light green area), curved runs (light blue area) and the stop and head-casts (light red area) behaviours as observed in larvae.

For example, associative learning is believed to be based on experience-driven plasticity of the mushroom body (MB) pathway (*Gerber et al., 2004*) and here learning is seen as a change in the effective signal transmission of this pathway, which runs parallel to the innate one, before both converge onto the motor system. The effective strength of signal transmission is changed by synaptic plasticity that may result in inverting the relative balance between the MB outputs that control attraction vs. aversion (as in the adult [*Aso et al., 2014*, *Owald et al., 2015*]) in an odour-specific

manner. After convergence with the other pathways, a net inversion would correspond to a change in the sign of the gain in our model (*Figure 3*).

This is consistent with the results of *Schleyer et al. (2015a)* showing that learning and changes in odour concentration yield similar motor effects; as well as the results of *Gepner et al. (2015)* suggesting that larvae combine olfactory and visual signals upstream of the decision to turn. It also explains the apparent similarity of taxis behaviours observed across modalities (compare for odour: *Gomez-Marin et al., 2011*; light: *Kane et al., 2013*, temperature: *Lahiri et al., 2011*, and why when aversive light stimulus and attractive olfactory stimulation are perfectly synchronized their antagonistic effects appear to blend smoothly (see *Bellmann et al., 2010*, Figure 1C). A state-based model, by contrast, needs to postulate that all these factors have equivalent and parallel effects on multiple mechanisms to account for these results (*Davies et al., 2015*).

## Integrating lateral oscillation with peristaltic motion

In the current work, we examined chemotaxis behaviour in the absence of a mechanism for stopping. The models' results show that peristalsis inhibition is not crucial for the emergence of the taxis signatures discussed above. However, it is clear that crawling speed in larvae can be reduced down to zero, and this is correlated with large body bends (*Figure 1* tail speed), and thus has an impact on chemotaxis behaviour. We here briefly speculate on how crawling speed might be included in our model in future.

An initial hypothesis is that stopping could be a by-product of making large turns, i.e., that the physical constraints of body bending lead to the peristaltic wave being disrupted. However, our analysis (consistent with *Gepner et al., 2015*; *Hernandez-Nunez et al., 2015*) shows that the inhibition of the forward motion is triggered on average at the onset of the turn and is thus not a mere consequence of large body bends (*Figure 9B*). We also note from our analysis that speed reduction is graded (*Figure 1B,C*), although contrary to turning amplitude (*Figure 2D*), the tail speed distribution is bimodal (*Figure 9A*). This is not necessarily indicative of a bimodal control signal ('decision to stop'); however, as such discontinuities (*Figure 9D*) can be explained by the non-linear dynamics arising from simple models of peristaltic wave propagation (*Ross et al., 2015*) as the speed is reduced. Our simple suggestion, then, is that the same signal that changes the oscillation amplitude could also directly change the peristalsis speed, which below some threshold results in a stop.

There is some experimental data suggesting that average speed, frequency of stopping events and average body-bending amplitude co-vary across stimuli conditions. Larva both reduce their crawling speed and increase their turn-rate in response to large step increases in $CO_2$ (*Gershow et al., 2012*). Larvae tend to accelerate when moving up a chemical gradient (*Gomez-Marin and Louis, 2014*) which is correlated with a reduced turn-rate (*Schleyer et al., 2015a*). Fast escape responses involve reduced head sweeping and rapid crawling (*Ohyama et al., 2015*). In some paradigms, crawling speed and turning amplitude are not so clearly inversely correlated, e.g., large step increases in ethyl acetate concentration decrease turn-rate but reveal only a weak non-significant increase in speed (*Gershow et al., 2012*); however, in the same paper, when the larvae move in a spatial gradient, the inverse correlation between speed and turn-rate is observed. In addition, peristalsis and head sweeps cease together under some conditions, e.g., sudden light offsets (*Kane et al., 2013*), but this may be a freezing reaction unrelated to normal taxis behaviour.

## Predictions

We have proposed a model in which re-orientation is always active during taxis, as part of an oscillatory sensorimotor program. This mechanism makes predictions that differ from taxis arising from the alternation of discrete-action motor programs (*Gomez-Marin et al., 2011*; *Gomez-Marin and Louis, 2014*; *Kane et al., 2013*), as implemented in alternative state-based models (*Davies et al., 2015*; *Schleyer et al., 2015b*; *Ohashi et al., 2014*).

We have assumed that all head-sweeps are under the control of the same oscillator, independently if the animal has stopped or is running. The head-sweep amplitude should decrease under favourable changes, such as moving up-gradient towards an attractive odour, and increase under unfavourable changes. It also follows that 'stop and head-cast' events should be preceded by a smaller head-sweep in the opposite direction, which indeed agrees with our analysis of larval tracks

in (*Figures 1B* and *2A,B*), whatever the conditions. This relationship implies that 'stop and head-cast' and 'weathervaning' behaviours are not independent mechanisms, and hence, for example, it should not be possible to obtain neurogenetic control over one without affecting the other.

The oscillator hypothesis predicts that stimulus elicited responses are timing dependent. Specifically, the continuous model predicts that experimentally varying the timing of stimulus exposure, for example via optogenetically induced olfactory stimulation, would trigger head-sweeps whose magnitude and direction vary as a function of the state of the oscillator at the time of stimulation. In contrast, a state-based model would predict that a step stimulus during re-orientation could trigger a termination of the head-sweep and a transition to the run state, thus 'accepting' a new heading direction. Thus, it does not predict a similar relationship between the stimulus timing, nor its strength, and the amplitude of the head-sweeps.

Our model also has potential consequences for learning. Current theory supports the view that memory expression is due to efficacy changes in synapses converging to mushroom body output neurons (MBONs) that encode valence; stimulating these neurons can elicit attraction or aversion (*Aso et al., 2014*; *Sachse and Beshel, 2016*). However, our model suggests that the mushroom body pathway is yet another pathway that converges to modify the effective gain in the motor system, and thus MBON valence would also be sensitive to activation timing. That is, it should not be possible to yield a change in odour attraction behaviour via optogenetic activation of a target MBON unless the stimulus respects the timing sensitivity of the larval oscillator, mimicking a closed-loop motor action-stimulation.

## Conclusion

Larval taxis behaviour has been characterised as transitions between discrete states, or actions (*Green et al., 1983*; *Sawin et al., 1994*; *Cobb, 1999*; *Gomez-Marin and Louis, 2012*) requiring 'action-selection' or 'decision-making' processes (*Gomez-Marin and Louis, 2014*). Here, we presented an alternative hypothesis according to which taxis results from a single simple sensory-motor process (*Figure 9C*): sensory signals directly modulate the continuous lateral oscillations of the anterior body, which we observed in larvae (*Figure 1*). Despite their simplicity, our models capture a remarkable number of taxis phenomena observed in larvae and suggest an elegant picture in which all types of sensory signals, mono or multi-modal, can combine by simply converging on the single process that lies at the core of taxis: a turning oscillator. Additional new features such as different sensory receptors or intermediate relays, such as the MB pathway, can be directly integrated, and their respective influence can be modulated by simply changing the intensity of their signals.

It has been argued that over long time scales, natural selection favours not merely effective innovations, but systems that flexibly enable the incorporation of innovations (*Vermeij, 1973*). The modularity of the system described here could provide such an evolutionary flexibility since it allows for behaviour to adapt by simply plugging in or removing input modalities. In future work, we will investigate the algorithmic nature of the proposed mechanism, how it operates within the complexity of larvae body mechanics and its robustness under increasing neural realism and additional processes required for the various tasks larvae perform in natural environments.

## Materials and methods

### Real larvae path analysis

We analysed the tracks from 42 wild-type larvae, the data recorded for *Gomez-Marin et al. (2011)*, which was supplied by Matthieu Louis. Each 3rd-instar foraging-larva path was recorded for 5 min at 7fps after releasing each larva on a rectangular agarose slab opposite to an odour source given by an ethyl butyrate droplet suspended from the lid. Tail, centroid, and head positions were extracted from each frame of the video using custom tracking software. Having obtained the processed data we used Matlab to analyse the tracks. Body bending was calculated as the angle formed between the tail-to-centroid axis and centroid-to-head axis. The variable 'angular velocity of the anterior part of the body' was obtained as the derivative across time of the centroid-to-head axis orientation. Specifics of the path analysis are presented where appropriate in the result sections.

## Agent-based simulation in discrete time steps

The agent model is an abstract description of the mechanism we believe larvae use to move up or down stimulus gradients. It consists of a point with position $x_n, y_n$ and an associated orientation $\theta_n \in -2\kappa\pi, +2\kappa\pi$. The model runs in discrete time $n \in \{1 \cdots N\}$, with each time-step representing an iteration of an algorithm. The agent's algorithm is simple, and we provide a MATLAB implementation for download (*Wystrach et al., 2016*). It is summarised by the following state-update equations that need to be executed in their order of appearance :

$$\theta_n \leftarrow \theta_{n-1} + H(\theta_B + g \times (s_T + p_{n-1}))(-1)^n \tag{1}$$
$$x_n \leftarrow x_{n-1} + \lambda \sin \theta_n \tag{2}$$
$$y_n \leftarrow y_{n-1} + \lambda \cos \theta_n \tag{3}$$
$$s_n \leftarrow C(x_n, y_n) \tag{4}$$
$$p_n \leftarrow s_n - s_{n-1} \tag{5}$$
$$n \leftarrow n+1 \tag{6}$$

assuming initial conditions for sensory input $s_0 = 0$ and a random initial position set for $x_0, y_0$ and orientation angle $\theta_0$.

At each time-step the agent moves distance $\lambda$, in the updated direction $\theta_n$, which depends on the intrinsic turning pattern, alternating left or right for odd or even time-steps. The baseline amplitude of the lateral oscillation is set to the baseline angle $\theta_B$. In the presence of environmental stimulation, the baseline angle is modified by sensory input $s_n$ determined by the concentration $C(x_n, y_n)$ at the current location. This includes a phasic signal $p_n = s_n - s_{n-1}$, which corresponds to the change of stimulus intensity perceived between two time-steps, and, optionally, a tonic signal $s_T$, which corresponds to the absolute stimulus intensity perceived at a given step. The strength of modulation depends on the gain $g$, and $H(x)$ is a hard-limit function:

$$H(x) = \begin{cases} x & \text{if } 0 \leq x \leq \pi \\ \pi & \text{if } x > \pi \\ 0 & \text{if } x < 0 \end{cases} \tag{7}$$

A negative gain $g$ with a positive $p_{n-1}$ (i.e. an increase in concentration perceived) on one step would lead to a decrease in turning away at the next step (up to the lower boundary of $H(x)$), while a negative $p_{n-1}$ (i.e. a decrease in concentration perceived) will lead to an increase in turning away on the next step (up to the higher boundary of $H(x)$). Thus, with a negative $g$, the resulting paths tend to be directed towards the odour source, while a positive $g$ would mediate repulsion.

The function $C(x, y)$ could be a fixed odour-gradient map, or a bivariate normal distribution (see *Equation 25*) that can be used to represent the distribution of odour concentration around an odour source. The maps of odour gradients used in our simulations have been provided by Matthieu Louis' Lab, as recorded in (*Gomez-Marin and Louis, 2014*). Stronger odour source concentrations were modelled by simply scaling the gradient map. If the agent hits the boundary of the odour gradient map, a new orientation is randomly assigned so the agent keeps within the boundaries.

In some conditions, we added noise (see results). The additive noise is modelled simply as :

$$\theta_n \leftarrow \theta_{n-1} + H(\theta_B + g \times (s_T + p_{n-1}))(-1)^n + Z_n, \tag{8}$$

where $Z_n$ is drawn from normal distribution and then added to the agent's current heading angle.

## Agent-based simulation in continuous time

The continuous agent is abstracted to an oriented point-sensor (as in the discrete-time model) but now, critically, the change in heading displays inertial moments and is driven by a model of coupled neural oscillators. The model attempts to capture the dynamics of heading change in continuous time, given that the stimuli are integrated by the driving non-linear oscillator and that re-orientation motion is constrained by some form of muscle-body constraints.

We use the single-segment model of the lamprey (*Lansner and Griller, 1997*) to represent the neural oscillator driving the change in heading of the agent. The CPG consists of a pair of compartments, here taken to driving changes in heading of the agent (*Figure 8A*). Each compartment

contains a pool of excitatory neurons $E$ and a cross-inhibitory interneuron $C$, which projects to the opposite compartment. The $E$ unit of *Figure 8A* with its self-connection, therefore, stands for the activity of a pool of excitatory neurons that interconnect within the compartment and project to the $C$ inhibitory neuron, while both $E$ and $C$ receive an inhibitory connection from the $C$ neuron of the opposite compartment. Further the $E$ neurons of both compartments receive input from the $A$ unit, which represents pooled sensory input, and a modulatory influence from the $S$ unit, whose effects will be described shortly.

Our model is based on the version of *Wilson (1999)* of the lamprey simplified model (*Lansner and Griller, 1997*) according to which the neuronal responses are given at the spike-rate level given by the *Naka and Rushton (1966)* function:

$$R(x,h) = \begin{cases} \frac{mx^n}{h^n+x^n} & \text{if } x \geq 0 \\ 0 & \text{if } x < 0 \end{cases}, \tag{9}$$

which maps the stimulus intensity $x$ of the net synaptic input to the expected spike-rate response of a neuron. The parameter $h$ sets the half-response threshold while $n$ sets the steepness of the response, which we take here to be $n = 2$. Spike-rates can only take positive values and therefore the function is constrained to lie in the positive integers up to the maximum $m$, which here will be set to $m = 100$ throughout. Each neuron also accounts for a spike-rate adaptation effect due to a slow after-hyperpolarization potential current $I_{AHP}$, which operates by raising the half-response threshold $h(t)$ of *Equation 9*. The equations for the left side of the coupled oscillators that we examined are as follows :

$$\tau\frac{dE_L}{dt} = -E_L + R(A + W_{ee}E_L - W_{ec}C_R, 64 + g(A)H_{EL}) \tag{10}$$

$$\frac{dH_{EL}}{dt} = \frac{1}{\tau_H(A)}(-H_{EL} + E_L) \tag{11}$$

$$\tau\frac{dC_L}{dt} = -C_L + R(A + W_{ce}E_L - W_{cc}C_R, 64 + g(A)H_{CL}) \tag{12}$$

$$\frac{dH_{CL}}{dt} = \frac{1}{\tau_H(A)}(-H_{CL} + E_L), \tag{13}$$

where $E_L, C_L$ represent the excitatory and cross-inhibitory neuron of the left compartment in *Figure 8A*, while the $H_X$ represents the dynamics of the $I_{AHP}$ of a neuron, $R(x,h)$ is the *Naka and Rushton (1966)* function of *Equation 9*, and $W_x$ are the synaptic weights shown in *Figure 8A*. On the same figure, we see that the neuromodulatory unit $S$ connects to both compartments, its effects are exerted via modifying the time constant and gain of the $I_{AHP}$:

$$g(A) = 6 + (0.09A)^2 \tag{14}$$

$$\tau_H(A) = \frac{35}{(1+0.04A^2)}, \tag{15}$$

where an increase in the input from $A$ will result in an increase of the $I_{AHP}$ gain and a decrease in its time constant $\tau_H$. The neural model has the respective equations for the right compartment, containing $E_R$ and $C_R$ for the right side oscillator:

$$\tau\frac{dE_R}{dt} = -E_R + R(A + W_{ee}E_R - W_{ec}C_L, 64 + g(A)H_{ER}) \tag{16}$$

$$\frac{dH_{ER}}{dt} = \frac{1}{\tau_H(A)}(-H_{ER} + E_R) \tag{17}$$

$$\tau\frac{dC_R}{dt} = -C_R + R(A + W_{ce}E_R - W_{cc}C_L, 64 + g(A)H_{CR}) \tag{18}$$

$$\frac{dH_{CR}}{dt} = \frac{1}{\tau_H(A)}(-H_{CR} + E_R), \tag{19}$$

To represent the biophysical constraints on re-orientation due to body-bending in real larvae, we used an idealized linear spring-mass-damper acting on the change in heading of the agent see *Figure 8B*. The system uses a pivoting spring-damper-mass on a joint to represent the elastic and damping forces exerted by the surrounding cuticle under the influence of opposing muscle forces

driving body-bending in the larva. The muscles would normally be driven by motor neurons, but here, we simplify by assuming that the motor neurons replicate the activity of the $E_L$ and $E_R$ premotor neurons and thus the later can be directly used.

The agent continuously moves at speed of 1 mm/s in the direction indicated by the body angle $\theta$. This simplification is justified in terms of our finding that the peristaltic wave peaks are uncorrelated with the body bending and thus can be taken as slow motion of the posterior body segment following the heading direction indicated. However, our model does not capture the straightening of the body bend due to this motion, or the friction forces exerted from the contact with the ground withholding the restoration. Given that the oscillation is driven by the premotor neuron activity and that the larva is assumed to continuously move at constant speed, the details of how the body bending is restored have been simplified out in our model to be driven by restorative elastic forces of the body. We take a non-dimensionalized approach writing the muscle model driving the head as second order system of idealized spring-mass-damper (see *Fung, 2013*):

$$\frac{d^2\theta(t)}{dt^2} = -2\zeta\theta'(t) - k\theta(t) + (E_L(t) - E_R(t)), \tag{20}$$

where $\zeta = \eta/(2\sqrt{k\gamma})$ defines the damping ratio, with $\eta$ the damping force coefficient, $k$ the stiffness coefficient of a linear spring and $\gamma$ the muscle gain. We assume muscles on each side of the body work against each other to change the heading and thus, in this two-dimensional model, the net torque produced is taken to be the difference in spike rates of the premotor neurons $E_L(t) - E_R(t)$ driving the muscles on each side. Evidently, the system is not representative of the larval muscle activity but the change in orientation caused by this activity. Nevertheless, it allows us to examine an embodied sensory-motor process during chemotaxis in continuous time avoiding the use of a detailed body that in essence would still only describe the motion of the olfactory sensor at the larva model which is needed for our demonstration. Further, writing the system in this form allows us to avoid having to consider specific values for the parameters and examine a generic system described by a level of damping, for which we have chosen an intermediate value $\zeta = 1/2$. The bearing $B$ is calculated via an integration of the change of heading angle $\theta$ in *Equation 20*. The continuous forward motion towards the current bearing is then converted to Cartesian coordinates to indicate the position of the head as a point:

$$\frac{dB}{dt} = \frac{\theta(t)}{10} \tag{21}$$

$$\frac{dx}{dt} = \sin\frac{B(t)}{10} \tag{22}$$

$$\frac{dy}{dt} = \cos\frac{B(t)}{10}, \tag{23}$$

the factors of 10 are simply used here to scale-down the motion of the agent so it looks similar to the scale used in the discrete agent model. Lastly, we define the $A$ neuron's activity pattern which we assumed to be representative of an olfactory sensory neuron. $A$'s output is a combination of a tonic output $b_T$, which is required to maintain the oscillation but also influences the oscillation frequency *Lansner and Griller (1997)*, along with the derivative of the odour concentration $C(t)$ superimposed :

$$A(t) = b_T + G\frac{dC}{dt}, \tag{24}$$

where $g$ defines the gain defining how much the derivative of the sensory stimuli alters the firing rate of input $A$, which perturbs the motor patterns and in turn influences the sensed stimulus in a closed-loop such that the rhythmic behaviour generates input for adaptive control (*Willis and Arbas, 1997a*). The sensory stimuli is drawn from a virtual odour gradient that is simply taken to be a scaled bivariate normal distribution:

$$M(x,y) = \frac{1}{2\pi\sigma_x\sigma_y\sqrt{1-\rho^2}}$$
$$\exp\left(-\frac{1}{2(1-\rho^2)}\left[\frac{(x-\mu_x)^2}{\sigma_x^2} + \frac{(y-\mu_y)^2}{\sigma_y^2} - \frac{2\rho(x-\mu_x)(y-\mu_y)}{\sigma_x\sigma_y}\right]\right) \tag{25}$$

with $\rho = \frac{\text{Cov}(x,y)}{\sigma_1 \sigma_2}$ being the correlation of $x$ and $y$. The sensory information as a function of time is then given by :

$$C(t) = cM(x(t), y(t)). \tag{26}$$

The model system was evaluated numerically, with the parameter set and initial conditions listed in *Table 1*, using mathematics software from Wolfram Research, Inc. (2015) (for an example file see *Wystrach et al., 2016*. For our purposes, the choice of parameters was broad and any arbitrary set that has sufficiently strong contralateral inhibition $W_{cc}$ such that the left-right oscillators quickly lock in antiphase while the frequency of the oscillation falls approximately within the larval range of 0.5 Hz, was sufficient.

Further, we examined the change of bearing in response to a step change in the input firing rate of $A$. For these results, input from the odour gradient was ignored and the change of bearing was examined in response to a step increase of amplitude $A_m$ in the input at various time points $t_s$:

$$A(t) = b_T + A_m U(t - t_s), \tag{27}$$

where $U(t)$ is the unit step function with an onset time at $t_s$. The change of bearing was measured by integrating the head angle for long enough time for it to settle back to its cycle of zero-average change of bearing. Each curve of *Figure 8F* consists of $10^2$ points covering $t_s$ timing over a full cycle

**Table 1.** CPG model parameter set and initial conditions. $M(x,y)$ is the multinomial distribution of *Equation 25*.

| Parameters | |
|---|---|
| $W_{cc}, W_{ec}$ | 4 |
| $W_{ce}$ | 1/10 |
| $W_{ee}$ | 3 |
| $b_T$ | 19 |
| $\tau$ | 1/10 |
| $m$ | $10^2$ |
| $n$ | 2 |
| $c$ | $10^3$ |
| $\rho$ | 1/5 |
| $\zeta$ | 1/2 |
| $k$ | 1 |
| **Initial Conditions** | |
| $C_L(t \leq 0), C_R(t \leq 0)$ | 0 |
| $E_L(t \leq 0)$ | 80 |
| $E_R(t \leq 0)$ | 20 |
| $H_{EL}(t \leq 0)$ | 0 |
| $H_{ER}(t \leq 0)$ | 0 |
| $H_{CL}(t \leq 0)$ | 0 |
| $H_{CR}(t \leq 0)$ | 0 |
| $\theta(t \leq 0)$ | 0 |
| $B(t \leq 0)$ | 0 |
| $A(t \leq 0)$ | $b_T$ |
| $g(t \leq 0)$ | $6 + (9A(0)/100)^2$ |
| $\tau_H(t \leq 0)$ | $35/(1 + 0.04A(0)^2)$ |
| $C(t \leq 0)$ | $M(x(0), y(0))$ |

of oscillation (ie. from one peak of $E_L$ burst to the next), while each curve differs in the step amplitude $A_m$.

## Obtaining the frequency spectrum of head velocities

We sample the heading speed of each larva trajectory at $\Delta t = 1/10$ and then perform a discrete Fourier transform of each of the heading speed vectors $\theta^j$ of trajectory $j$ :

$$F^j_{s,r} = \frac{1}{\sqrt{n}} \sum_{r=1}^{n} \exp \left[ 2\pi i \frac{(s-1)(r-1)}{n} \right] \theta^j_r.$$

(28)

The spectrum plot shown in *Figure 8C* represents the mean spectrum out from the speed vectors of $n = 25$ trajectories. The starting point of each trajectory is distributed according to a squared matrix of points centred on the odour source (see starting positions in *Figure 8D*). Each trajectory's initial point condition $x(0), y(0)$ is set to point on this square matrix with a horizontal and vertical distance of 10mm between each point.

## Acknowledgements

We are grateful to Matthieu Louis for providing us with larval tracking data and for his feedback on our manuscript. We would also like to thank Daniel Malagarriga for his comments and corrections on our manuscript, Bertram Gerber and Michael Schleyer for the useful discussions. This work was supported by the EU FET-Open grant MINIMAL.

## Additional information

### Funding

| Funder | Grant reference number | Author |
| --- | --- | --- |
| Seventh Framework Programme | FP7-618045 | Antoine Wystrach Konstantinos Lagogiannis |

The funders had no role in study design, data collection and interpretation, or the decision to submit the work for publication.

### Author contributions

AW, Conception, Design and analysis of discrete agent model, Analysis of tracking data, Conception and design, Analysis and interpretation of data, Drafting or revising the article; KL, Conception, Design and analysis of discrete and continuous agent model, Conception and design, Analysis and interpretation of data, Drafting or revising the article; BW, Supervision of project, Including input into interpretation of data and design of work, Drafting or revising the article

### Author ORCIDs

Konstantinos Lagogiannis, http://orcid.org/0000-0001-9349-801X

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
