## [Decision Letter]

Thank you for submitting your article "Continuous lateral oscillations as a core mechanism for taxis in *Drosophila* larvae" for consideration by *eLife*. Your article has been reviewed by two peer reviewers, and the evaluation has been overseen by K VijayRaghavan as the Senior Editor and Reviewing Editor. The following individuals involved in review of your submission have agreed to reveal their identity: Eduardo J Izquierdo (Reviewer #2).

The reviewers have discussed the reviews with one another and the Reviewing Editor has drafted this decision to help you prepare a revised submission.

Summary:

Wystrach et al. present a simple model for navigation and show that simulations of this model produce trajectories that are consistent with those observed in some measurements of larval *Drosophila* chemotaxis. The paper is clear and well-written, and it is relatively easy to follow the authors' main arguments. The paper provides support for the hypothesis that taxis behavior in *Drosophila* larva is the result of modulation of a single behavior (i.e., a continuous lateral oscillation in the anterior body). Some claims are overstated and there needs to be some work on the Discussion (see below).

The authors provide experimental evidence through analysis of the larva's body bends showing the presence of continuous lateral oscillation. They then use an idealized computational model to test the hypothesis that modulation of such oscillation in the head is sufficient to reproduce the different orientation strategies observed in the larva (weathervaning, klinotaxis, and klinokinesis). This result is significant because larval taxis has been typically characterized as transitions between multiple behaviors, and this computational model shows a proof of concept that such an assumption is not necessary – and that a simpler strategy is sufficient to account for the observed behavior. The authors present a second model that is continuous in time, to further validate their hypothesis. One of the interesting insights is that the descending sensory signals that control the behavior do not need to be stronger on the different sides of the body. This result echoes the model presented by Izquierdo and Lockery, 2009, as the authors acknowledge.

Strengths: Overall, the goal, assumptions and insights from the model are rather well presented and easy to follow. Both a discrete and continuous time model are presented. Both make testable predictions that can be compared to existing or easily acquired data. The authors also explain some quantitative measures made in other experiments through the lens of their model.

Essential revisions:

Weaknesses and important suggestions for improvement:

The paper could do a better job of enumerating and consolidating these predictions and especially that the authors would emphasize where their model makes different predictions from existing models of taxis. Instead the emphasis seems to be on showing that their model can produce agreement with existing data. There are two problems with this approach: first, just because a simulation can reproduce an experimental result does not prove the simulation to be correct. Second, the authors have made some odd choices of experimental results with which to compare their model, with emphasis seeming to be on making very broad claims about the model's power, rather than stringently testing its powers of predictions (see technical discussion). Some attention needs to be paid to the citations, which are incomplete and sometimes off-target.

The authors should be clearer how it is that the model addresses klinokinesis. It is not clear how modulation of the oscillatory pattern can modulate the probability to run/stop. Also, typically klinokinesis involves random reorientation of the heading – is this also accounted for by the modulation of the oscillatory pattern?

One minor issue with the paper is that it might be lengthier than necessary. While *eLife* does not have a length limit, the authors could do one more pass at the writing to find suitable places to make the wording more succinct. The Discussion section is particularly lengthy.

[Editors' note: further revisions were requested prior to acceptance, as described below.]

Thank you for resubmitting your work entitled "Continuous lateral oscillations as a core mechanism for taxis in *Drosophila* larvae" for further consideration at *eLife*. Your revised article has been favorably evaluated by K VijayRaghavan as Senior Editor, a Reviewing Editor, and one reviewer.

The manuscript has been improved but there are some remaining issues that need to be addressed before acceptance, as outlined below:

Many of these could have been addressed as they were clearly mentioned in the first set of reviews.

While we appreciate that the authors have made an effort towards improving this paper, we cannot recommend that it be published in *eLife* as currently written, because it still contains significant flaws that were not corrected in revision. The authors still need to do the following:

a) Fix notation.

\nabla s is universally understood to be the gradient of s – a vector of the spatial derivatives of s. Writing (section “4.2 Agent-based simulation in discrete time steps”) "\nabla s_{n-1} = s_{n-1} – s_{n-2} represents the change in the sampled stimulus between time steps n-1 and n-2" is to equate this vector with a scalar difference in the value at two time points. Were it to be published in this form, it would cause great confusion and possibly cause readers to question the mathematical competency of the authors. It is astounding that after this basic error was pointed out gently to the authors, they failed to correct it.

[Disp-formula equ1] on the same section is open to misinterpretation, because of confusing notation – H(…) is a function, but g(…) represents multiplication by a scalar. The authors put a note below, but it would me much more effective to place a symbol (e.g. g*(…)) to indicate multiplication.

b) Shorten the manuscript, focusing on significant new results and testable predictions that differentiate the model from existing ones.

The strengths of this paper are first the novelty of the suggestion that small continuous lateral head movements and larger "head-sweeps" or "head-casts" are in fact the same basic behavior and that modulation of the amplitude of these oscillations is the sole computation underlying navigational behaviors and second the presentation of two numerical model that embody this suggestion. This hypothesis leads to a number of testable predictions about the movements of larvae performing chemotaxis, and publication of this work will likely provoke interesting experimental work to test these predictions. This is why I support publication of the paper in *eLife*.

As an example of the testable predictions that I hope would be emphasized in a revised paper, consider lines 210-214: The agent tends to spend more time with the odour on its sides. Is this true only for radially symmetric odorant distributions created by diffusion from droplets (in which case this may be a geometric effect), or in a linear gradient would we expect that larvae would also be more likely to orient at angles to the direction of highest concentration? Run and tumble models would predict that in a linear gradient, the most probable orientation would be directly towards higher concentration, so this is a significant prediction that differentiates this model from others.

The second paragraph of the section “4.2 Agent-based simulation in discrete time steps” details another prediction that can test the model: if you provide a sharp stimulus change at a particular point in the oscillatory cycle, you should evoke a particular pattern of movement that is different from what would be predicted by the head-sweep acceptance model. I think this prediction could be extended or amplified – does it imply a resonant frequency at which an applied stimulus would evoke the greatest changes?

The third paragraph of the section “4.2 Agent-based simulation in discrete time steps” on the other hand does not present a test of the model: although lateralized sensing is not required for the model to function, it does not follow that the larva cannot have a lateralized internal representation of body posture, any more than the fact that the model does not include cessation of forward movement means that the larva cannot stop. So being able to condition a larva to turn left would not falsify the model; conversely the failure of lateralized operant conditioning could indeed be due to a lack of lateralized internal representation but also to a number of other causes (to my knowledge, no one has demonstrated that larvae can be operantly conditioned in any context). It is also not clear that the competing "run and tumble" model based on state transitions requires a lateralized internal representation.

c) Greatly reduce discussion of effects of changing gain and eliminate speculation about circuit function.

The simple model presented here – response to odor is linear in odor change with rectification and saturation – is not a complete description of chemotaxis. For instance, invariant response to odor over orders of magnitude concentration requires that the gain be a function of concentration, but this is not incorporated in mathematical description of the model in section 4. Further, current research indicates that olfactory receptor neurons do not respond linearly to odor concentrations (or to changes in odor concentration) and that behavioral responses to olfactory neuron activity are also nonlinear. It's completely fine to present a simplified model, especially if it makes it easier to understand the main features, but too much of the paper then focuses on the trivial results of adjusting these simplified model parameters.

Any model of olfaction necessarily has a "gain," a conversion from odor concentration (measured in number or mass per volume) to sensory neuron activity (measured in spikes per second, membrane voltage, inward current, etc.). For reduced models that do not attempt to represent the full neural pathway, the "gain" may instead represent a transformation from odor concentration to activity in some downstream neuron or a more generalized neural representation of odor quality and quantity. In the case, as presented here, that the gain simply multiplies the input stimulus, inverting the gain is mathematically equivalent to inverting the input stimulus, with the effect that taxis towards high concentration becomes taxis towards low concentration and vice versa. It is a completely trivial result that a switch from attraction to aversion can be effected by inverting the gain; to understand learning the interesting research question, not addressed in this work, is how this inversion is accomplished, especially in an odor-specific manner.

To be clear, "gain" is a model fit parameter. It has a well-defined meaning only in the context of section 4.2, [Disp-formula equ1] and section 4.3 [Disp-formula equ9]. Discussion about learning changing gain, olfactory receptor neurons summing gain, motivation or habituation modifying gain, etc. are simply repeating the fact that changing this model parameter changes the behavior of the model. The authors should go back over their manuscript and remove (or shorten to at most a few sentences of discussion in total) any part that discusses the effect of changing the gain, unless they also present a testable, quantitative, and mechanistic explanation of how this change is accomplished.

The authors should also remove the discussion of neuroanatomical circuit elements. Nothing in the model as written directly represents any part of the larva's nervous system (even the continuous time model uses a simplified CPG model adapted from lamprey, with no evidence that such a system actually exists in *Drosophila* larvae). Attempting to place abstract and generalized model features into specific circuit contexts only leads to confusion and contradiction.

As an example, in section 3.5, it is proposed that ORNs respond with an ORN-odor-specific gain and that these gains are summed together to produce the overall response to an odor. (This is not entirely consistent with the existing understanding of *Drosophila*'s olfactory system, which includes mutual inhibition mediated by local interneurons in the (larval) antennal lobe.) In section 2.6, learning is modeled as a change in gain. Taken together, 2.6 and 3.5 imply learning results from changes in responses of ORNs to odors. This contradicts both the current consensus of the field and the discussion in the later parts of section 3, which place learning at the mushroom bodies.

We recommend that section(s) 2.5 be shortened significantly, 2.6-2.8 eliminated entirely, 3.5 and 3.6 eliminated entirely, and 3.8 refocused on testable behavioral predictions, not on circuit architecture. In the rest of the manuscript, the authors should take the opportunity presented by another round of revision to consider what is most essential about this work and shorten the manuscript so that these important features are not lost. The Discussion currently occupies 10 manuscript pages (1/3 of the paper excluding Methods). I would recommend that it be shortened to at the very most 5 pages, and preferably 3.

---

## [Author Response]

*[…] Essential revisions:*

*Weaknesses and important suggestions for improvement:*

*The paper could do a better job of enumerating and consolidating these predictions and especially that the authors would emphasize where their model makes different predictions from existing models of taxis. Instead the emphasis seems to be on showing that their model can produce agreement with existing data. There are two problems with this approach: first, just because a simulation can reproduce an experimental result does not prove the simulation to be correct.*

We have now explicitly consolidated the predictions of our model in a new section of the discussion dedicated to predictions (3.8), which also attempts a comparison to previous models.

To our knowledge the only explicit simulation models of larval chemotaxis published to date are Schleyer et al. 2015, Davies et al. 2015 and Ohashi et al. 2014. All three have in common that they can be formulated as a markovian state transition model, assuming a switch between a turning (re- orientation) state, during which the larva is stopped, and a run state. However, only the Davies et al. 2015 gives a mechanistic account of how chemotaxis operates (state transition probabilities being functions of recent sensory input which is dependent on movement of a sensor through the gradient).

The Schleyer et al. 2015 and Ohashi et al. 2014 models instead are using an agent to either evaluate if measured variables can explain changes in chemotaxis performance (e.g. in Schleyer et al. 2015 the agent's head- turn angle is randomly chosen by sampling an empirical distribution at a region defined by bearing-to-odour and distance from odour source) or to examine the contribution of weathervaning (biased runs) to chemotaxis performance. As such, direct comparison of predictions is difficult.

Nonetheless, with this new prediction section, we believe it is now clear that our model does not just confirm existing results but provide new and unexpected predictions, some of which we could actually test through our new analysis of the data presented here.

*Second, the authors have made some odd choices of experimental results with which to compare their model, with emphasis seeming to be on making very broad claims about the model's power, rather than stringently testing its powers of predictions (see technical discussion).*

We have now clarified our choice of experimental results, which include both specific predictions, and exploration of the broader explanatory power, which we think is particularly important given the simplicity of the underlying concept. See further response under technical discussion.

*Some attention needs to be paid to the citations, which are incomplete and sometimes off-target.*

Indeed, we have added Ohashi et al. 2014, Gershow et al. 2012 in relevance to the Davies et al. 2015 model, and also the work on temperature gradients (Klein et al., 2014; Luo et al. 2010) as suggested. Also, Asahina et al. 2009 and Gomez-Marin et al. 2011 were added to the discussion on ORN saturation and the conversion to repulsion.

*The authors should be clearer how it is that the model addresses klinokinesis. It is not clear how modulation of the oscillatory pattern can modulate the probability to run/stop. Also, typically klinokinesis involves random reorientation of the heading – is this also accounted for by the modulation of the oscillatory pattern?*

The models presented in this paper do not include (do not address) klinokinesis (stopping). This was a deliberate decision as our goal is to understand the extent to which the oscillatory mechanism alone could produce chemotaxis phenomena, and we have now clarified this point. Nevertheless, it is reasonable to ask how stopping behaviour, which is apparent in larval chemotaxis, could be integrated into the model in future. As such, in the Discussion (3.7) we tentatively suggest a mechanism for stopping (Figure 12D) that could combine with oscillation. We have now revised this section of the Discussion to clarify the extent to which this mechanism is, and is not, consistent with existing experimental data (we included the suggested citations) and what it would predict.

*One minor issue with the paper is that it might be lengthier than necessary. While eLife does not have a length limit, the authors could do one more pass at the writing to find suitable places to make the wording more succinct. The Discussion section is particularly lengthy.*

We have tried to improve the wording to make the paper more concise.

[Editors' note: further revisions were requested prior to acceptance, as described below.]

*The manuscript has been improved but there are some remaining issues that need to be addressed before acceptance, as outlined below:*

*Many of these could have been addressed as they were clearly mentioned in the first set of reviews.*

*While we appreciate that the authors have made an effort towards improving this paper, we cannot recommend that it be published in eLife as currently written, because it still contains significant flaws that were not corrected in revision. The authors still need to do the following:*

*a) Fix notation.*

*\nabla s is universally understood to be the gradient of s – a vector of the spatial derivatives of s. Writing (section “4.2 Agent-based simulation in discrete time steps”) "\nabla s_{n-1} = s_{n-1} – s_{n-2} represents the change in the sampled stimulus between time steps n-1 and n-2" is to equate this vector with a scalar difference in the value at two time points. Were it to be published in this form, it would cause great confusion and possibly cause readers to question the mathematical competency of the authors. It is astounding that after this basic error was pointed out gently to the authors, they failed to correct it.*

We now appreciate that this use of \nabla may be confusing to some readers and we have removed it from the manuscript and modified Figure 3 and Figure 9 accordingly.

Our use of \nabla was referring to the backwards operator as used in discrete mathematics and the mathematics of finite differences to denote differences: \nabla f(t) = f(t) – f(t-dt), or \nabla x_n = x_n – x_{n-1} (see [1]), whereas the

operator \Delta is meant to denote forward differences \Delta s_n = s_{n+1} – s_n. Therefore, we believe using \Delta may also be confusing to some readers as this would require us to adjust the indexing of the algorithm to read\Delta s_{n-2} at step n (i.e. a value from two steps back).

In order to avoid any unnecessary confusion, we have decided to define a new variable p_n = s_n – s_{n-1}, and added a step to the algorithm to assign this new variable.

*[Disp-formula equ1] on the same section is open to misinterpretation, because of confusing notation – H(…) is a function, but g(…) represents multiplication by a scalar. The authors put a note below, but it would me much more effective to place a symbol (e.g. g*(…)) to indicate multiplication.*

We have introduced the \times symbol to clarify multiplication.

*b) Shorten the manuscript, focusing on significant new results and testable predictions that differentiate the model from existing ones.*

*The strengths of this paper are first the novelty of the suggestion that small continuous lateral head movements and larger "head-sweeps" or "head-casts" are in fact the same basic behavior and that modulation of the amplitude of these oscillations is the sole computation underlying navigational behaviors and second the presentation of two numerical model that embody this suggestion. This hypothesis leads to a number of testable predictions about the movements of larvae performing chemotaxis, and publication of this work will likely provoke interesting experimental work to test these predictions. This is why I support publication of the paper in eLife.*

*As an example of the testable predictions that I hope would be emphasized in a revised paper, consider lines 210-214: The agent tends to spend more time with the odour on its sides. Is this true only for radially symmetric odorant distributions created by diffusion from droplets (in which case this may be a geometric effect), or in a linear gradient would we expect that larvae would also be more likely to orient at angles to the direction of highest concentration? Run and tumble models would predict that in a linear gradient, the most probable orientation would be directly towards higher concentration, so this is a significant prediction that differentiates this model from others.*

Indeed, such experiments would be ideal, as they would allow a clear cut and an accessible way to discriminate models. Given our agent's behaviour in gradients is rather predictable however, we believe the example proposed would not be able to discriminate between models. Our agent stops re-orienting when the change in sensory stimuli is balanced between the left and the right part of the oscillation. Thus, it will re-orient itself towards the high-gradient, in essence operating as a gradient ascent algorithm tracking the odour-peak. In this sense it is indeed the radial distributions that give rise to the orbital behaviour and the agent spending more time with the odour on its side.

*The second paragraph of the section “4.2 Agent-based simulation in discrete time steps” details another prediction that can test the model: if you provide a sharp stimulus change at a particular point in the oscillatory cycle, you should evoke a particular pattern of movement that is different from what would be predicted by the head-sweep acceptance model. I think this prediction could be extended or amplified – does it imply a resonant frequency at which an applied stimulus would evoke the greatest changes?*

Correctly so, here we point to the model's general prediction that responses would be stimulus-timing depend against the oscillation phase. This is a general property arising from the coupled neural oscillator implementation; according to which stimuli perturb the oscillator dynamics and the response will depend on state of the oscillator at the time of stimulation.

However, the neurons we have used in this implementation are simplified spike-rate models and we expect that introducing spiking neural models, either phenomenological or biologically inspired ones would produce richer and more complicated stimulus responses. Thus, the idea that there may be particular stimulation patterns that resonate with the dynamics to amplify head-sweeps is very interesting, however we believe such predictions should be deferred for when some of the neural dynamics of the oscillator (which we suggest could be located in the VNC) have been identified so we can produce richer models and more specific predictions on stimulus-response.

*The third paragraph of the section “4.2 Agent-based simulation in discrete time steps” on the other hand does not present a test of the model: although lateralized sensing is not required for the model to function, it does not follow that the larva cannot have a lateralized internal representation of body posture, any more than the fact that the model does not include cessation of forward movement means that the larva cannot stop. So being able to condition a larva to turn left would not falsify the model; conversely the failure of lateralized operant conditioning could indeed be due to a lack of lateralized internal representation but also to a number of other causes (to my knowledge, no one has demonstrated that larvae can be operantly conditioned in any context). It is also not clear that the competing "run and tumble" model based on state transitions requires a lateralized internal representation.*

We have removed this paragraph.

*c) Greatly reduce discussion of effects of changing gain and eliminate speculation about circuit function.*

*The simple model presented here – response to odor is linear in odor change with rectification and saturation – is not a complete description of chemotaxis. For instance, invariant response to odor over orders of magnitude concentration requires that the gain be a function of concentration, but this is not incorporated in mathematical description of the model in section 4. Further, current research indicates that olfactory receptor neurons do not respond linearly to odor concentrations (or to changes in odor concentration) and that behavioral responses to olfactory neuron activity are also nonlinear. It's completely fine to present a simplified model, especially if it makes it easier to understand the main features, but too much of the paper then focuses on the trivial results of adjusting these simplified model parameters.*

The sections examining and/ or discussing the effects of gain change have been significantly reduced (see later comments on removal of sections).

*Any model of olfaction necessarily has a "gain," a conversion from odor concentration (measured in number or mass per volume) to sensory neuron activity (measured in spikes per second, membrane voltage, inward current, etc.). For reduced models that do not attempt to represent the full neural pathway, the "gain" may instead represent a transformation from odor concentration to activity in some downstream neuron or a more generalized neural representation of odor quality and quantity. In the case, as presented here, that the gain simply multiplies the input stimulus, inverting the gain is mathematically equivalent to inverting the input stimulus, with the effect that taxis towards high concentration becomes taxis towards low concentration and vice versa. It is a completely trivial result that a switch from attraction to aversion can be effected by inverting the gain; to understand learning the interesting research question, not addressed in this work, is how this inversion is accomplished, especially in an odor-specific manner.*

In section 3.2 we clarify our view of how odour-specific gain inversion can be implemented in learning. In short, it is synaptic plasticity from Kenyon Cells (KCs) to mushroom body output neurons (MBONs) that can modify the effectiveness of the MB pathway to relay sensory signals, in an odour-specific way. A change in the sign of the gain corresponds to altering the balance between signal transmission via attractive and aversive MBONs enough such that, after convergence with the other pathways, the sign of the net sum is also inverted leading to inverting motor system responses to sensory stimuli (aversion vs. attraction).

*To be clear, "gain" is a model fit parameter. It has a well-defined meaning only in the context of section 4.2, [Disp-formula equ1] and section 4.3 [Disp-formula equ9]. Discussion about learning changing gain, olfactory receptor neurons summing gain, motivation or habituation modifying gain, etc. are simply repeating the fact that changing this model parameter changes the behavior of the model. The authors should go back over their manuscript and remove (or shorten to at most a few sentences of discussion in total) any part that discusses the effect of changing the gain, unless they also present a testable, quantitative, and mechanistic explanation of how this change is accomplished.*

We agree that the gain is only a model parameter, and we have clarified this in the manuscript: “This ‘gain’ is only a model parameter and does not represent any specific neuro-anatomical feature in larvae. Rather it could be seen as the net sum of all pathways from the stimulation of sensory receptors to muscle contraction.” (see section 3.2, which has now been shortened).

In contrast to other models we think what is important to stress here is that this model requires only one degree of freedom to capture a substantial range of characteristics of the animal's behaviour, under various conditions and this is potentially significant for understanding the underlying mechanism of chemotaxis.

The crucial conceptual point is that this variety of effects may simply "sum up". This idea, if correct, has strong consequences for the underlying circuits.

We now try to convey this idea succinctly in section 3.2 (simple sensory mapping), and transferred the gist of the results into a single result section 2.5 and associated Figure 5.

Note that we removed entirely the story about fructose, phasic and tonic connections (ex-sections 2.8 and 3.4).

*The authors should also remove the discussion of neuroanatomical circuit elements. Nothing in the model as written directly represents any part of the larva's nervous system (even the continuous time model uses a simplified CPG model adapted from lamprey, with no evidence that such a system actually exists in Drosophila larvae). Attempting to place abstract and generalized model features into specific circuit contexts only leads to confusion and contradiction.*

*As an example, in section 3.5, it is proposed that ORNs respond with an ORN-odor-specific gain and that these gains are summed together to produce the overall response to an odor. (This is not entirely consistent with the existing understanding of Drosophila's olfactory system, which includes mutual inhibition mediated by local interneurons in the (larval) antennal lobe.) In section 2.6, learning is modeled as a change in gain. Taken together, 2.6 and 3.5 imply learning results from changes in responses of ORNs to odors. This contradicts both the current consensus of the field and the discussion in the later parts of section 3, which place learning at the mushroom bodies.*

We have now removed most instances and clarified the connection between the model and 'circuit speculation' to prevent confusion. Much of it was already gone with the deleted sections (see above). However, as we argued above, we believe that discussion of neuroanatomical circuits should not be removed entirely, as it may yield specific predictions for neuro- biologists; for example, in our predictions we have the following sentence (which we were asked to elaborate on): “…our model suggests that the mushroom body pathway is yet another pathway that converges to modify the effective gain in the motor system, and thus MBON valence would also be sensitive to activation timing.”

We also kept a short Discussion in section 3.1 on evidence towards a CPG in the ventral nerve cord and how lateral bending is linked to cross-connections, which serves to materialize the nature of our oscillator.

Otherwise we removed the general discussion about OSN, CPGs, as well as the subesophageal zone as a potential candidate for the convergence of the multiple sensory inputs.

*We recommend that section(s) 2.5 be shortened significantly, 2.6-2.8 eliminated entirely, 3.5 and 3.6 eliminated entirely, and 3.8 refocused on testable behavioral predictions, not on circuit architecture. In the rest of the manuscript, the authors should take the opportunity presented by another round of revision to consider what is most essential about this work and shorten the manuscript so that these important features are not lost. The Discussion currently occupies 10 manuscript pages (1/3 of the paper excluding Methods). I would recommend that it be shortened to at the very most 5 pages, and preferably 3.*

We removed as suggested, the sections 2.5 2.6, 2.7, 2.8 (and their respective figures) and also the respective Discussion ex-section 3.4,

3.5 and 3.6, and changed the section title of 2.5 to encompass discussion on change of gain which we rewrote for terseness. Our Discussion was 3300 words is now 1800 words.